# Personality intervention affects emotional stability and extraversion similarly in older and younger adults

Gabriela Küchler[1], Kira S. A. Borgdorf[2,3], Corina Aguilar-Raab[2,4], Wiebke Bleidorn[5], Jenny Wagner [6] & Cornelia Wrzus [1,7] ✉

Past research showed that personality traits develop less strongly after younger adulthood, though the underlying processes remain poorly understood, and personality intervention studies scarcely investigated age differences. Also, existing findings are mostly limited to explicit assessments of personality traits (i.e., questionnaires). In this preregistered, multi-method study, we examined associations between changes in personality states and explicit and implicit trait self-concepts of emotional stability and extraversion throughout an 8-week socio-emotional intervention, 3 and 12 months later. The sample consisted of younger and older adults (N = 165, age range = 19-78 years). Findings indicate changes in personality states, explicit self-concepts for both traits, and the implicit self-concept of extraversion. Only state changes in emotional stability predicted changes in the corresponding explicit but not implicit trait self-concept. Importantly, the effects were consistent across age groups, and exploratory analyses showed higher engagement among older adults throughout the intervention. The findings emphasize that older adults might benefit as much from socio-emotional interventions as younger adults, and potential age differences in skill acquisition might be set off through engagement.

Many people desire to change some of their personality characteristics, such as becoming more outgoing or better at handling stress[1]. Yet, mere intention is often insufficient, emphasizing the need for effective interventions[2–4]. Evidence suggests that personality interventions can elicit targeted changes in self-reported traits, yet other trait manifestations and the underlying processes remain underexamined[2,4,5]. Theoretical models highlight gradual shifts in personality states and self-reflections as mechanisms of long-term trait change[5–7]. Although most studies focused on young adults[2], meta-analytic work demonstrates that personality development continues throughout adulthood, yet is more pronounced in younger adults[8–10], raising questions about reduced trait malleability in late adulthood[5].

In the present research, we addressed these gaps and examined (a) whether short-term personality state changes lead to lasting trait changes during and beyond a socio-emotional intervention and (b) how these changes differ between younger and older adults. Furthermore, we expanded past operationalizations, which primarily relied on self-report questionnaires assessing explicit self-concepts, by including indirect measures

capturing implicit representations of self-concepts—focusing on the Big Five traits of emotional stability and extraversion, which are most often desired to change[4].

Processes theories on personality development highlight pre-action, action, and post-action factors that may facilitate or hinder personality trait change[5–7,11]. Pre-action factors include the desire and belief in feasible change, while action factors involve engaging in relevant behaviors, such as reserved individuals acting more outgoing than usual. Post-action factors include self-reflections and attributions of changed behaviors. In this research, we focus on personality states as action factors and examine how their changes during an intervention might subsequently influence explicit and implicit trait self-concepts of emotional stability and extraversion. Pre- and post-action factors were addressed through goal setting, discrepancy awareness, and reflection.

Personality states represent momentary feelings, thoughts, and behaviors related to trait-domains[12] and are considered situation-dependent manifestations of personality traits[6,13,14]. At the same time, trait-congruent

¹Department of Psychological Aging Research, Psychological Institute, Heidelberg University, Heidelberg, Germany. ²Clinical Psychology, Interaction- and Psychotherapy Research, Faculty of Social Sciences, University of Mannheim, Mannheim, Germany. ³Heidelberg University, Heidelberg, Germany. ⁴Institute of Medical Psychology, University Hospital Heidelberg, Heidelberg, Germany. ⁵Univerity Zurich, Zurich, Switzerland. ⁶University Hamburg, Hamburg, Germany. ⁷Network for Aging Research, Heidelberg University, Heidelberg, Germany. ✉e-mail: wrzus@psychologie.uni-heidelberg.de

states presumably reinforce stability, while incongruent states promote trait changes under certain conditions[5]. States are often assessed momentarily via experience sampling (e.g., refs. 5,15), or over short periods of time, such as the past day or week to capture short-term expressions of traits[16,17]. State emotional stability reflects reactions to difficult situations and emotional intensity and stability[15,18,19]. State extraversion involves social and assertive behavior, positive affect, and high energy in social situations[20,21].

Consistent with dual-process models of personality, we conceptualize personality traits being mentally represented in two systems, explicit and implicit self-concepts[5,13,22]. The explicit self-concept represents consciously accessible self-evaluations, typically measured via questionnaires. In contrast, the implicit self-concept comprises rather automatic associations between the self and personality characteristics, accessible through indirect measures such as the Implicit Association Test (IAT)[23]. Previous research found moderate links between explicit and implicit self-concepts[13,24], likely due to different accessibility, methodological factors, and also developmental pathways[5,6,25].

States could contribute to changes in explicit and implicit traits through different pathways: Changes in automatic components shape implicit self-concepts through associative processes like reinforcement learning[5,26]. For example, repeated positive feedback for staying calm under pressure may strengthen the implicit "me–calm" association tied to emotional stability. In addition, deliberate aspects of states correspond to the explicit self-concept and presumably change through reflective processes such as self-reflection. Reflection can (a) incorporate repeated experiences into the explicit self-concept (e.g., interpreting calmness as increased stability), and (b) support self-regulation by bridging gaps between actual and desired self via novel states (e.g., applying stress management techniques[5,6,13,26].

First longitudinal studies found that repeated extraverted states predicted changes in explicit self-concepts of extraversion[21,27]. Similarly, repeated stress-related negative affect predicted decreases in explicit self-concepts of emotional stability[17,27,28]. One previous study has linked states to changes in implicit self-concepts, showing effects for extraversion but no other Big Five traits[27].

Although many people desire to increase emotional stability and extraversion, this often does not result in trait changes (for review see ref. 2). One reason for these findings may be that changing personality traits might be difficult, particularly in adulthood, when traits are comparatively stable[8,10,29]. For lasting changes, it appears essential to use strategies that initiate and sustain changes in states and related self-concepts[2,5,7,30].

Recent meta-analyses further support the effectiveness of interventions in changing personality traits[2]. These interventions have used strategies of three broad categories: (a) motivators such as goal setting and awareness of discrepancy between current and desired trait levels, (b) behavioral practice, (c) self-reflection to target beliefs, insight, and the reinforcement of new behaviors[2,31,32]. These strategies operate across pre-action, action, and post-action stages, with reflective processes presumably shaping mainly explicit self-concepts, while associative processes, such as reinforcement and feedback learning, shape mainly implicit self-concepts, and behavioral practice contributes to both explicit and implicit self-concepts[5,32].

Despite these insights and interventions, empirical studies rarely examined state changes and reflection directly[2,33]. Also, research on how states and traits may change jointly during interventions remains limited[34]. Initial evidence suggests state changes can precede trait changes for resilience[35] and mindfulness[36]. However, most studies have relied on self-report questionnaires of explicit self-concepts. These measures capture self-evaluations but not indirect representations such as implicit self-concepts[13]. Moreover, compared to indirect measures, explicit self-reports tend to be more susceptible to social desirability and demand effects[37]. Such effects are particularly relevant in interventions when participants desire trait changes. Addressing these gaps with multi-method approaches is crucial for understanding the nuanced processes that contribute to personality change, which then enables the development of effective interventions.

Previous studies on volitional personality development have predominantly focused on younger samples, limiting the generalizability to

older individuals. Moreover, studying older adults could clarify whether slower personality development[8,38] reflects differences in processes of personality development. We propose that older adults may exhibit less pronounced state changes (i.e., thoughts, feelings, behaviors) due to slower associative[39] and reinforcement learning[40]. Furthermore, with age, state changes could affect personality self-concepts less strongly. This may result from reduced self-reflection[41] and a focus on integrating experiences into existing self-concepts[42]. For example, older adults less often compare themselves to others or their past selves regarding personality traits and other characteristics[43]. Also, older adults favor identity assimilation, thus preserving consistency in their self-views[42]. Disruptions in learning processes may likewise hinder modifications in implicit self-concepts through state changes[39].

So far, only a few studies have examined age differences in state-trait links directly, showing mixed findings. One observational study found that changes in daily stress reactivity were more strongly associated with changes in trait emotional stability among younger adults compared to older adults[15], while another study observed no significant age differences in state-trait links[27]. Findings from clinical and personality interventions are mixed, with some indicating less effectiveness among older adults[44–47], and others showing no substantial age differences[48–51]. Multiple factors, such as age differences in analyzed samples, learning, motivation, or adherence, could explain these mixed results. Overall, theory and some evidence suggest that younger samples may benefit more from interventions, but age differences in state-trait links have yet to be examined in controlled interventions.

The present research has two aims: First, expanding the scarce literature on personality state-trait associations, we aimed to investigate whether intervention-based state changes in emotional stability and extraversion would predict changes in trait self-concepts. We assessed personality traits via self-reports (i.e., explicit self-concepts) and employed indirect measures of implicit self-concepts for a more comprehensive assessment. Second, we aimed to provide insights into age differences in state and trait changes and their associations.

Personality traits were measured before (T1), four weeks into (T2), and after the intervention (T3), as well as at three (T4) and twelve months (T5) follow-ups; implicit measures were not assessed at T4. Personality states were assessed as weekly averages throughout the intervention.

While the term state is commonly used in experience sampling studies to describe situation-specific experiences[27], here we focus on week-to-week changes in states, capturing short-term expressions compared to general trait assessments. Although weekly averages of states are less temporally fine-grained than momentary assessments, they remain distinct from trait self-concepts, as they reflect recent behavioral patterns rather than generalized self-evaluations. For example, participants rate how they behaved during the past week (e.g., "stressed" vs. "relaxed"). In contrast, traits refer to more stable self-concepts, which are assessed as general evaluations without a specific time frame (e.g., "I am someone who is relaxed and handles stress well"). Thus, weekly assessments remain consistent with theoretical conceptualizations of personality states as behaviorally anchored short-term constructs.

We conducted an 8-week in-person intervention grounded in principles of personality development and change interventions[5,7,31]: State changes in emotional stability and extraversion were facilitated through psychoeducation, behavioral instructions, and coping strategies. Self-reflection targeted explicit self-concepts by raising awareness of discrepancies between current and desired trait levels and reflecting on state changes (see detailed intervention overview in Supplementary Table S1). Sessions 1–4 focused on emotional stability and emotion regulation, while sessions 4–8, addressed extraversion and interpersonal competencies.

We hypothesized that state emotional stability (H1a) and state extraversion (H1b) would improve continuously. We further expected that the improvement of state extraversion would be more pronounced during the second half of the intervention, which focused on social competencies. Further, we expected that the intervention would lead to increases in the explicit self-concept of emotional stability (H2a) and extraversion (H2b), as

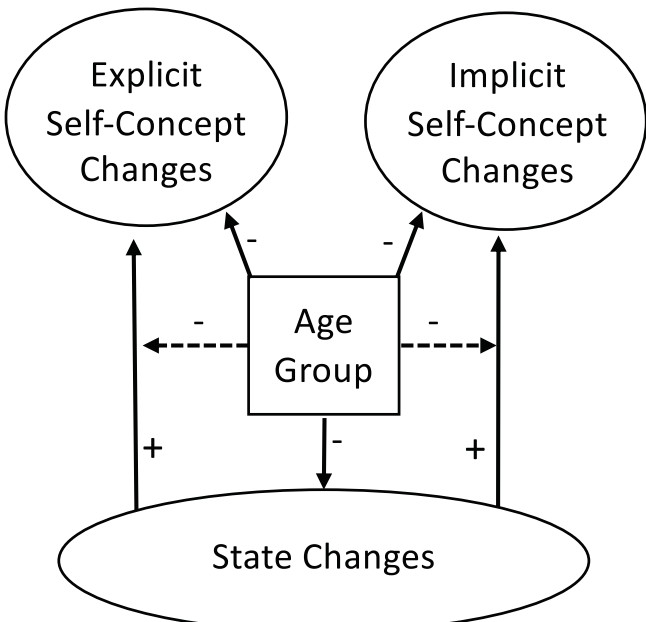

**Fig. 1 | Conceptual model of hypothesized relationships among state changes, explicit and implicit self-concept changes, and age group.** Solid arrows represent hypothesized direct effects; dashed arrows represent moderation effects. "+" and "−" signs indicate expected direction of associations. Age Group was coded "0" for younger adults and "1" for older adults.

well as the implicit self-concepts of both traits (H3a and H3b, respectively), with more pronounced increases in explicit than implicit trait self-concepts (H4). Consistent with process theories of personality change[5,6,11], we expected that more pronounced changes in state emotional well-being would be associated with stronger changes in trait emotional stability (H5a) and more pronounced changes in state social behavior with stronger changes in trait extraversion (H5b).

Regarding age differences, we hypothesized that the increases in state emotional stability (H6a) and extraversion (H6b) would be more pronounced among younger compared to older adults. Also, we expected increases in explicit self-concepts and implicit self-concepts to be more pronounced among younger adults compared to older adults (H7a, H7b). Lastly, we hypothesized that the association between changes in state emotional well-being and trait emotional stability (H8a), and between social behavior and extraversion (H8b) would be stronger in younger than older adults.

Exploratorily, we investigated the long-term effects of the intervention over 3 and 12 months after the intervention, and age differences in intervention engagement. These analyses were not pre-registered. Figure 1 depicts a conceptual model of the hypothesized relationships between state changes, trait changes, and age.

## Methods
### Transparency and openness
We report how we determined our sample size, data exclusions, and all measures in this study. The study design, sampling rationale, hypotheses, and data analyses were preregistered on January 16, 2023, OSF:
1) https://osf.io/bhwpx
2) https://osf.io/c7p8y

At the time of the preregistration, several participants were screened for eligibility (see below for the criteria) and had answered the initial baseline questionnaire (T1). At that point, the intervention had not started, and no longitudinal data had been collected. We did not conduct any data inspection before the preregistration. Minor deviations from the pre-registration are detailed in Supplementary Table S2. The hypotheses were

rephrased to improve grammatical clarity and consistency. The anonymized data, code, and materials are publicly accessible on OSF. The study was approved by the Ethics Committee of Heidelberg University, and all participants provided informed consent before participation. The study was performed in line with the principles of the Declaration of Helsinki, and we followed JARS[52] for reporting standards.

This is the second manuscript submitted from the data of this project. The first manuscript focused on the validation of the intervention and changes in mindfulness, self-compassion, and related constructs[53].

### The socio-emotional intervention
The intervention was in-person and aimed at promoting emotional stability, extraversion, and mental health in healthy adults. It was conducted over 8 weeks in 2-h sessions. Materials were available in paper-pencil format and online as audio resources. Participants who missed sessions could access materials online and attend catch-up sessions.

Each group (five to twelve participants) was composed of the same participants across the intervention and guided by two trainers to facilitate a familiar, secure social environment. Groups included both younger and older adults. Groups met on separate weekdays and sessions took place in the course rooms at the Heidelberg University and University Hospital in Heidelberg, Germany.

Part 1 addressed stress, resilience, attention, and emotion regulation, aiming primarily to increase state and trait emotional stability (see Fig. 2). Part 2 emphasized interpersonal socio-emotional competencies, including social dynamics, systemic perspectives on social interactions, practicing social skills with video feedback, and summarizing the training content. Thus, the last four sessions aimed to increase state and trait extraversion. Nevertheless, exercises for emotional stability were still encouraged in weekly tasks. Like in other validated programs (e.g., Cognitively-Based Compassion Training[54]), this approach encompasses participants regulating their own attention and emotions first, before addressing interpersonal conflicts. Participants selected a training buddy in the first session and received daily tasks, including emotion regulation exercises, written self-reflections, and planned social interactions, to facilitate engagement in between weekly sessions. We provide a detailed outline of each session in Supplementary Table S1.

Overall, the intervention design integrated methods from evidence-based programs, including Mindfulness-Based Stress Reduction[55], Cognitively-Based Compassion Training[54], the Social Emotional Ethical Learning program[56], Acceptance and Commitment Therapy[57,58], systemic counseling[59], and Group Training of Social Competences (GSK)[60]. The intervention was tailored for the specific purpose of this study by the project leader, Corina Aguilar-Raab, who is a licensed psychotherapist, mindfulness, and compassion trainer, and Kira Borgdorf, who is a licensed systemic counselor. Additionally, the training was discussed and modified within the project group, who are experts in the research on personality development.

Twenty-seven graduate students in psychology and educational sciences were trained over 3 months, earning course credits. Their training involved a semester-long train-the-trainer seminar, combining participation in the intervention with subsequent instruction in learning didactical principles, such as maintaining a trainer's attitude and managing challenging situations. This initial trial also served as a pilot test, enabling minor session adjustments (e.g., exercise duration). Like the participants, the trainers were informed about the study's objectives but remained blind to specific hypotheses (e.g., age differences). All trainers were younger adults. Previous research suggests that age discrepancies between therapist and client do not weaken the therapeutic alliance and may even strengthen it in the case of younger therapists[61]. Further, overall effects of the therapist-client alliance on outcomes are small in group therapy settings[62].

Each trainer participated in at least two supervision sessions per training cohort. Like the seminar, these supervision sessions were led by the two project principals, Corina Aguilar-Raab and Cornelia Wrzus, who have extensive experience in teaching and training facilitation. Trainers reported

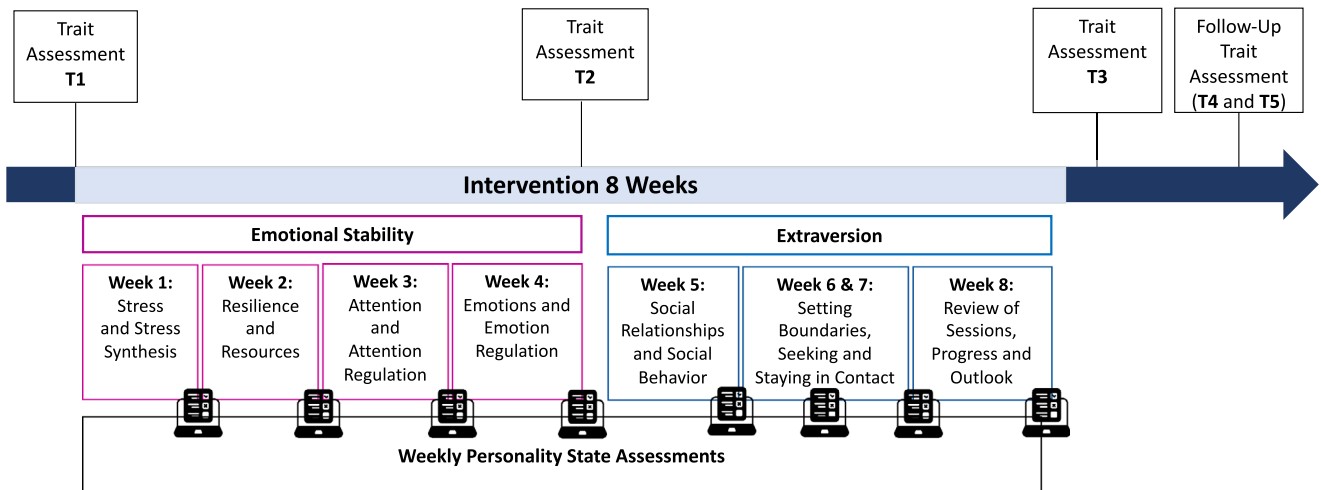

**Fig. 2 | Overview of assessments and intervention sessions.** T1–T3 represent trait assessments before, during, and after the 8-week intervention, while T4 and T5 indicate follow-up assessments conducted 3 and 12 months post-intervention. The intervention comprised eight weekly sessions, with weeks 1–4 (pink-framed) adhering closely to the training manual ($M = 5.80$, $SD = 1.14$ on a 7-point scale, where 7 indicated complete adherence).

targeting emotional stability and weeks 5–8 (blue-framed) targeting extraversion. Weekly personality state assessments were administered throughout the intervention.

**Recruitment of participants and screening procedure.** Based on power analysis, we aimed to recruit 220 participants in two age groups: 110 younger adults (18–35 + 2 years), and 110 older adults (aged 55–80 ±2 years). Power estimation used $1–\beta = 0.80$, $\alpha = 0.05$, targeting medium to large effect sizes while considering the feasibility of the in-person study. The intervention was a randomized controlled trial (RCT), with a waitlist control group. A separate publication by the project team validated the effectiveness of the training[53]. Since comparisons between the intervention group and the waitlist control group are limited to pre- and post-intervention changes and do not extend to the within-person processes examined in this manuscript, the RCT design is not further analyzed in the present study.

Recruitment used various online and offline channels (e.g., public lectures and advertising, flyers, social media) to reach a diverse sample. Advertisements described the training and study content, procedure, timeline, costs, and compensation. A weblink led to the online screening.

We used the online software SoSci Survey[63] for data collection. After informed consent for screening, participants underwent eligibility screening. Final criteria included: (1) age ≥18 years, between 18 and 35 ( + 2 years) or 50+ years, (2) internet access and suitable hardware, (3) no concurrent socio-emotional, compassion, or mindfulness training, or psychotherapy, (4) sufficient language skills, and (5) values below depression and anxiety cut-offs (PHQ-9[64], GAD-7[65];. People reporting suicide ideation or scoring above the cut-offs received online resources for therapeutic support. Eligible participants received a study ID and an information package on data privacy, study details, procedures (e.g., random group assignment), timeline, costs, and compensation. Based on evidence that enrollment fees improve adherence (e.g., refs. [66,67]), we charged a fee of 80 EUR (50 EUR reduced fee for individuals with low income, e.g., students/pensioners). Participants received a reimbursement of up to 110 EUR (completing all questionnaires) and a 50% fee refund when attending ≥5 sessions. They were informed about random assignment to intervention or waitlist control groups and enrolled online after informed consent.

In total, 1019 individuals completed the screening. Of these, 45% ($n = 458$) were excluded based on predefined criteria, such as elevated scores on the mental health screening or ongoing psychotherapy/other mental health trainings. Of those eligible, 63.7% ($n = 357$, 75% female, 25% male, 59% younger age group) did not enroll in the training. Overall, 203

individuals (19.9% of screened people) enrolled, of which 20 dropped out before the training started, and 18 did not complete the training (i.e., ≤ 4 training sessions). Enrollment occurred in three cohorts in January, April, and June 2023. Drop-out analyses showed no substantial differences between completers and dropouts (Supplementary Table S3). Supplementary Fig. S1 details the study and attrition flow.

**Final sample of participants in the personality intervention.** The final sample ($N = 165$) that participated in the intervention consisted of 80 younger adults ($M_{Age} = 28.33$, $SD_{Age} = 4.92$, range$_{Age}$ = 19–42, 75% female, 24% male, 1% non-binary, 65% university degree) and 85 older adults ($M_{Age} = 63.55$, $SD_{Age} = 7.20$, range$_{Age}$ = 50–78, 75% female, 25% male, 60% university degree). Because of difficulties in recruiting participants, we expanded the older age group to include individuals aged 50 years and older, and we included six participants whose ages fell between the predefined ranges for younger adults (18–35 years) and older adults (50+ years). The latter were categorized as younger adults, as their ages were closer to the mean and range of this group. More detailed socio-demographic information is displayed in Supplementary Table S4. Participants self-reported their gender/sex. We used the German term "Geschlecht", which does not specify whether responses should reflect gender identity or biological sex. This variable was reported for descriptive purposes only and was not included as a covariate, as there is no established theoretical or empirical basis for gender or sex differences in the context of personality change interventions. Individuals who participated at T4 ($N = 123$) and/or T5 ($N = 100$) did not differ significantly from participants who did left the study after T3, in any variable of interest or demographically. Table Supplementary S3 provides a detailed overview.

**Measures**
All assessments were conducted online using the SosciSurvey platform[63].

**Personality traits.** Explicit personality self-concepts were assessed as general evaluations without a specific time frame using the Big Five Inventory-2[68,69] (e.g., "I am someone who is relaxed and handles stress well."). Emotional stability and extraversion were measured with 12 items each. Participants rated each item on a 5-point Likert scale ranging from 1 (*disagree strongly*) to 5 (*agree strongly*). Since the original scoring key[69] was designed for negative emotionality/neuroticism, we reversed the coding direction. Both scales had very good reliability: Emotional stability $\omega = 0.85$–$0.89$, and extraversion $\omega = 0.86$–$0.88$.

**Table 1 | Descriptive statistics of explicit and implicit self-concepts of emotional stability and extraversion by age group**

| Variable *M (SD)* *n* | T1 | T2 | T3 | T4 | T5 |
|---|---|---|---|---|---|
| **Younger adults** | | | | | |
| **Emotional stability** | | | | | |
| Explicit | 2.86 (0.56) 80 | 3.08 (0.62) 74 | 3.22 (0.60) 68 | 3.28 (0.60) 62 | 3.24 (0.59) 48 |
| Implicit | 0.31 (0.36) 73 | 0.32 (0.38) 69 | 0.26 (0.46) 63 | | 0.25 (0.48) 46 |
| **Extraversion** | | | | | |
| Explicit | 3.21 (0.67) 80 | 3.36 (0.64) 74 | 3.44 (0.59) 68 | 3.41 (0.46) 62 | 3.29 (0.49) 48 |
| Implicit | −0.22 (0.49) 73 | −0.06 (0.51) 69 | −0.04 (0.46) 63 | | −0.02 (0.50) 45 |
| **Older adults** | | | | | |
| **Emotional Stability** | | | | | |
| Explicit | 2.85 (0.65) 81 | 3.06 (0.66) 77 | 3.28 (0.64) 73 | 3.36 (0.63) 61 | 3.35 (0.74) 52 |
| Implicit | 0.36 (0.37) 70 | 0.36 (0.35) 72 | 0.29 (0.36) 72 | | 0.45 (0.33) 48 |
| **Extraversion** | | | | | |
| Explicit | 3.21 (0.60) 81 | 3.31 (0.64) 77 | 3.37 (0.60) 73 | 3.30 (0.55) 61 | 3.29 (0.59) 52 |
| Implicit | −0.20 (0.62) 69 | −0.13 (0.60) 72 | −0.10 (0.63) 72 | | −0.02 (0.73) 48 |

The IAT assessed implicit self-concepts of emotional stability and extraversion[70]. The IAT is a reaction-time measure of the strength of rather automatic associations between self-concepts and attributes. Participants quickly sort words representing high or low trait levels, associating them with themselves or others. Faster reaction times when pairing a trait level with themselves compared to others represent a stronger implicit association with that specific trait level[70]. Prior research validated the IAT for measuring implicit attitudes and personality traits[13,23].

Specifically, the test included three practice blocks of 20 trials each (blocks 1, 2, 4) and two test blocks (blocks 3 and 5), with 60 trials per block for each trait[23,71]. Target categories (*me* and *others*) each included five stimuli (e.g., *I, myself, their, your*). Attribute categories (traits) included 5 stimuli each, contrasting anxiety versus calmness (e.g., *calm*) for emotional stability, and extraversion versus introversion (e.g., *talkative*) for extraversion. In test blocks 3 and 5, target and attribute stimuli were interchanged. The word order was randomized across blocks, and stimuli repeated without replacement until all trials were completed.

Implicit self-concept scores were calculated using built-in error penalties and winsorized reaction times[23,71] (i.e., <300 ms and > 10,000ms). Split-half reliabilities were acceptable: 0.70–0.76 for emotional stability, and 0.89–0.95 for extraversion. Descriptives statistics are displayed in Table 1. Correlations among all of personality traits measurements are displayed in Supplementary Table S5.

**Personality states.** Participants reported their personality states during the past week using six bipolar items for emotional stability (e.g., *stressed* versus *relaxed*) and four items for extraversion (e.g., *shy* versus *talkative*). The items were adapted from the Multidimensional Mood Questionnaire[72]. Participants responded on a scale ranging from 1 to 7. Internal consistency was excellent, with an average of $\omega = 0.92$ for state emotional stability and an average of $\omega = 0.88$ for state extraversion. The intraclass correlations (ICC) for emotional stability ICC = 0.38 and extraversion ICC = 0.47 show substantial variation in states within individuals across the intervention.

**Control variables.** Engagement in the intervention was assessed with 5 items in each weekly protocol. Participants rated how many weekly tasks they completed on a scale from 1 (*not at all*) to 7 (*completely*) and rated their use of audio material. Additionally, they reported how often they applied training knowledge and skills in daily life, both on a scale from 1 (*not at all*) to 7 (*daily*). Finally, participants rated the extent of interaction with their training buddy on a scale from 1 (*not at all*) to 5 (*intensively*).

Context factors were measured by three items. Participants rated on a bipolar scale how hectic (1 = *very hectic* to 7 = *very calm; reversed*), atypical (1 = *very untypical* to 7 = *very normal; reversed*), and how exhausting (1 = *very exhausting* to 7 = *very relaxed; reversed*) their week was.

**Analytic strategy**
We winsorized the values of all variables ($M \pm 3\,SD$) in cases of outliers ($n = 7$). We observed a small number of missing values, with an average of 2.3% for data from weekly protocols and 8.9% for trait measures. Age group was coded as 0 = younger adults (18–42 years) and 1 = older adults (50+ years), based on the bimodal distribution of the variable, and grand-mean centered. We used RStudio Version 1.4.1106 for data preparation and control analyses[73]. We tested normality and homoscedasticity, which showed only minor deviations and were considered negligible given the use of robust maximum likelihood estimator (MLR) and Bayesian estimation methods[74]. Further, we tested measurement invariance for each trait and type of self-concept, and strong measurement invariance held in each measurement model[75] (see Supplementary Table S6 for details).

For hypothesis testing, we conducted multilevel analyses to test our hypotheses regarding state changes (H1a, H1b, H6a, and H6b). At Level 1, we modeled time (a continuous variable coded from 0 to 7) as a random within-person effect. At Level 2, we included the age group as a fixed between-person effect. Further, we included cross-level interaction terms between time and age group as predictors. Additionally, to examine whether changes in extraversion were more pronounced in Part 2 of the training, we specified multilevel models incorporating a discontinuous dummy-coded time variable (Part 1 coded as 0, Part 2 coded as 1) and an interaction term between time (i.e., week) and intervention part (i.e., dummy variable). Statistical tests were two-sided. Effects were considered significant if $p < 0.05$ and the confidence interval excluded zero.

To test hypotheses involving trait change, we applied latent neighbor change (H2a-H4, H7a, and H7b; as used in ref. 15 and latent growth analyses[76] (H5a, H5b, H8a, and H8b) in Mplus Version 8.6[77]. Personality traits were modeled as latent variables. Latent explicit self-concepts were represented using three content-based parcels with averages of four items each, capturing the three facets of neuroticism/emotional stability (anxiety, depression, emotional volatility) and extraversion (*sociability, assertiveness, energy level*), respectively[69,78]. Latent implicit self-concepts were modeled with two parcels based on split-half D2 scores[70]. Neuroticism scores were reversed to represent emotional stability. We tested this model for each type of trait and self-concept separately.

To control for item-specific method variance, we included indicator-specific method factors (IS2 and IS3) for the second and third trait parcels, respectively (Fig. 2). These factors captured residual covariation among parcels across time points. They were specified with fixed loadings of 1 (i.e., constrained for identification) and constrained to be uncorrelated with the trait and growth factors, ensuring that only method-specific variance—rather than trait-related variance—was modeled. This modeling strategy is often preferred over specifying correlated residuals because it provides a more parsimonious and psychometrically robust approach to controlling for item-specific method effects[79].

As no baseline differences in age were observed in personality states and traits, we did not include age group as a predictor of intercept factors. However, age group was included as a predictor of slope factors to test whether the magnitude of change differed across age groups.

Model A was a latent neighbor change model estimating trait change across two neighboring time intervals (T1 to T2 and from T2 to T3). This model tested whether changes differed between the first and second intervention phases, while accounting for their temporal proximity and dependence (Fig. 3A). We used this approach to first examine a finer temporal resolution of change, which was especially relevant for extraversion because it was targeted in the second phase of the intervention (i.e., training sessions 5 to 8, corresponding to the time between T2 to T3). Age group was included as a predictor of trait change during each phase. We used the MLR with robust standard errors. Model fit indices of all latent neighbor change models were good and are displayed in Supplementary Table S7.

Model B was a bivariate latent growth model designed to test whether state changes predicted trait changes continuously during the intervention (see Fig. 3B[77]). Each model included an intercept and a growth factor. The intercept factor loadings were set to 1 across time, meaning the intercept represents the baseline level. Latent slope loadings were defined such that each unit increment represented one week. For the trait slope, loadings were set to 0, 4, and 8, corresponding to the approximately 4-week intervals between the three trait assessments (T1, T2, and T3). For the state slope, loadings were set to 0 through 7 across the eight weekly state assessments. The trait slope was predicted by the state slope, age group, and their interaction. Model fit indices of all latent growth models were good and are displayed in Supplementary Table S8.

Model C was a piecewise latent growth model used to examine long-term trajectories of trait change following the intervention, meaning whether changes were maintained, amplified, or diminished. For that purpose, change was modeled separately for the intervention phase (T1 to T3) and the follow-up phase (i.e., T3 to T5). Accordingly, we specified one intercept factor with factor loadings set to 1 across time, and two latent growth factors (see Fig. 3C): Slope 1 captured change across the intervention phase and a plateau thereafter, while Slope 2 captured potential change during the follow-up period. Slope loadings were specified as 3 and 12 to represent intervals of 3 and 12 months post-intervention. Both slopes were predicted by age group. Model fit indices of all piecewise latent growth models were good and are displayed in Supplementary Table S9. The model for explicit self-concept of extraversion showed good fit without age but demonstrated poor fit when age was included.

We used the Bayes estimator with default, non-informative priors[74] for Models B and C, which did not converge with MLR estimation likely due to the complexity. For Bayes estimation, we employed 10,000 iterations per analysis for explicit self-concepts and 20,000 for implicit self-concepts and models including latent interactions to achieve convergence with values below 1.1 of the Gelman–Rubin diagnostic (Potential Scale Reduction Factor, PSRF[74,80]). To verify estimation accuracy, we used the first half of iterations as a burn-in to ensure that estimates and PSRF values remained consistent when doubling iterations. The analyses provided point estimates and 95% credibility intervals (CI) for the posterior distribution, with effects deemed significant if the CI excluded zero.

### Reporting summary
Further information on research design is available in the Nature Portfolio Reporting Summary linked to this article.

## Results
Descriptives for the trait and state measures are presented in Tables 1 and 2, respectively, separately for each age group.

### Changes in personality states and trait self-concepts
Results on changes in personality states and traits before and throughout the intervention are displayed in Fig. 4. State changes were analyzed using multilevel analyses. As predicted (H1a and H1b), state emotional stability and extraversion increased ($b = 0.09$, 95% CI [0.06, 0.13], $p < 0.001$) and $b = 0.06$, 95% CI [0.03, 0.08], $p < 0.001$). Other than expected, state extraversion did not increase more during the second part of the intervention ($b = 0.03$, 95% CI [−0.30, 0.35], $p = 0.145$). More detailed statistics are provided in Supplementary Tables S10 and S11.

Focusing on trait changes, we employed latent neighborhood change analyses to model trait changes from T1 to T2 (Part 1) and T2 to T3 (Part 2) separately. Results indicated that, in line with our predictions (H2a and H2b), the explicit self-concepts of both emotional stability (Part 1: $b = 0.321$, 95% CI [0.229, 0.412], $p < 0.001$; Part 2: $b = 0.319$, 95% CI [0.222, 0.416], $p < 0.001$) and extraversion (Part 1: $b = 0.161$, 95% CI [0.098, 0.225], $p < 0.001$; Part 2: $b = 0.072$, 95% CI [0.004, 0.140], $p = 0.037$) increased during the intervention (also see Supplementary Table S12).

Contrary to H3a, the implicit self-concept of emotional stability did not increase during the intervention (Part 1: $b = 0.002$, 95% CI [−0.059, 0.063], $p = 0.944$; Part 2: $b = −0.049$, 95% CI [−0.105, 0.006], $p = 0.082$). In line with H3b, the implicit self-concept of extraversion increased substantially during Part 1 ($b = 0.098$, 95% CI [0.024, 0.172], $p = 0.009$) and showed a similar trend during Part 2 ($b = 0.062$, 95% CI [0.000, 0.124], $p = 0.051$), without reaching conventional levels of significance.

Results partially supported H4: Changes in the explicit self-concept were more pronounced regarding emotional stability ($b = 0.321$, 95% CI [0.229, 0.412], $p < 0.001$ vs. $b = 0.002$, 95% CI [−0.059, 0.063], $p = 0.944$). However, regarding extraversion, overlapping confidence intervals of change estimates indicate comparable changes for both types of self-concepts ($b = 0.161$, 95% CI [0.098, 0.225], $p < 0.001$ vs. $b = 0.098$, 95% CI [0.024, 0.172], $p = 0.009$; also see Supplementary Table S12).

### Associations in state-trait changes
Results partially supported H5a regarding the associations between states and traits: The more state emotional stability increased during the intervention, the more the explicit trait self-concept of emotional stability increased ($b = 0.367$, 95% CI [0.191, 0.674]), but there was no credible evidence that the implicit trait self-concept increased ($b = 0.018$, 95% CI [−0.059, 0.117]). Contrary to our assumptions (H5b), there was no credible evidence for associations between changes in state extraversion and changes in explicit trait self-concepts ($b = 0.010$, 95% CI [−0.239, 0.290]) or implicit self-concept of extraversion ($b = 0.010$, 95% CI [−0.942, 0.299]) throughout the intervention. For complete model results, see Supplementary Table S13.

### Age differences in changes of personality states, trait self-concepts, and state-trait associations
Contrary to our predictions of more pronounced intervention effects in young adulthood (H6a and H6b), age did not moderate state changes in emotional stability ($b = −0.01$, 95% CI [−0.08, 0.05], $p = 0.249$) and extraversion ($b = −0.01$, 95% CI [−0.06, 0.03], $p = 0.205$). Also, the analyses did not support age differences in changes in explicit self-concepts of emotional stability (H7a; Part 1: $b = 0.042$, 95% CI [−0.109, 0.192], $p = 0.586$; Part 2: $b = 0.092$, 95% CI [−0.037, 0.221], $p = 0.162$) or implicit trait self-concepts of emotional stability (H7b; Part 1: $b = 0.033$, 95% CI [−0.068, 0.134], $p = 0.520$; Part 2: $b = −0.008$, 95% CI [−0.116, 0.100], $p = 0.887$).

Similarly, regarding extraversion, the analyses did not support age differences in changes in explicit self-concepts (H7a; Part 1: $b = −0.004$, 95% CI [−0.123, 0.116]; Part 2: $b = 0.049$, 95% CI [−0.056, 0.154], $p = 0.359$) or implicit trait self-concepts (H7b; Part 1: $b = −0.090$, 95% CI [−0.220, 0.040], $p = 0.174$; Part 2: $b = 0.024$, 95% CI [−0.083, 0.132], $p = 0.657$).

Contrary to H8a and H8b, there was no credible evidence that associations in state-trait changes were stronger in younger adults compared to older adults. Specifically, there was no credible evidence that age group moderated the relationship between state and trait changes in explicit emotional stability ($b = −0.148$, 95% CI [−0.414, 0.092]) or extraversion ($b = −0.029$, 95% CI [−0.392, 0.322]). Similarly, there was no credible evidence that age group moderated the relationship between state and implicit trait changes in emotional stability ($b = 0.039$, 95% CI [−0.139,

**Fig. 3 | Latent neighbor change and growth models. A** Latent neighbor change model: Estimates latent trait change separately for each intervention phase and predicts trait change by age group. **B** Latent Growth Model: Examines trait changes across the intervention: Latent slopes of trait changes from T1 to T3 are predicted by the latent slope of state changes across the eight-week intervention, age group, and their interaction. **C** Piecewise Latent Growth Model: Estimates latent growth in personality traits across the intervention ($S_1$) and follow-up assessments ($S_2$). I Intercept, S Slope. Latent variables are represented in rounded forms. Latent traits were estimated with three indicators P (i.e., parcels), for explicit trait measures and two indicators for implicit trait measures at each measurement point. Measurement invariance was established by constraining intercepts and factor loadings to be equal for each measurement. Method effects over time were addressed using indicator-specific method factors (IS2, IS3).

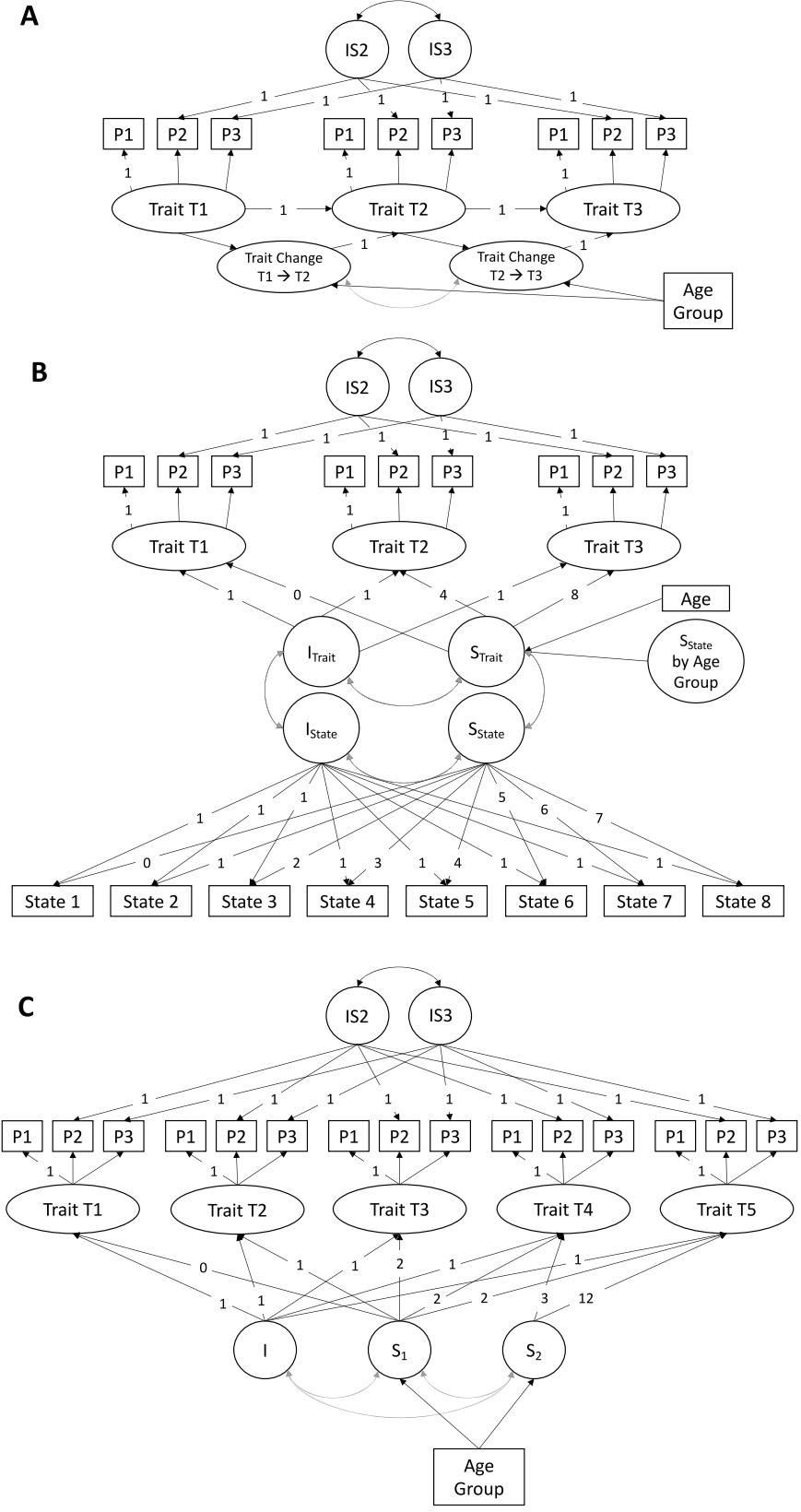

0.224]) or extraversion ($b = -0.116$, 95% CI [−2.033, 0.450], see Supplementary Tables S12 and S13 for more details).

To examine the evidence supporting the null hypothesis—meaning the true absence of age differences in the observed effects—we compared differences in the Bayesian Information Criterion (BIC) between models with and without age group as a predictor (see Supplementary Table S14, Model B, for details). For explicit emotional stability (BIC difference = 3.290), explicit extraversion (BIC difference = 4.609), implicit emotional stability (BIC difference = 5.209), and implicit extraversion (BIC difference = 4.199), BIC differences provided positive evidence for the null hypothesis,

**Table 2 | Descriptive statistics of state emotional stability and extraversion by age group**

| Variable M (SD) n | Week 1 | Week 2 | Week 3 | Week 4 | Week 5 | Week 6 | Week 7 | Week 8 |
|---|---|---|---|---|---|---|---|---|
| Younger adults | | | | | | | | |
| Emotional stability | 4.04 (1.20) 76 | 4.21 (1.00) 73 | 4.08 (1.43) 72 | 4.33 (1.70) 72 | 4.47 (1.63) 70 | 4.59 (1.56) 66 | 4.79 (1.71) 67 | 4.66 (1.77) 63 |
| Extraversion | 4.56 (1.25) 76 | 4.70 (1.00) 73 | 4.76 (1.09) 72 | 4.82 (1.40) 72 | 4.95 (1.29) 70 | 4.93 (1.42) 66 | 5.06 (1.14) 67 | 5.04 (1.05) 63 |
| Older adults | | | | | | | | |
| Emotional stability | 4.26 (1.51) 78 | 4.27 (1.68) 81 | 4.46 (1.63) 79 | 4.28 (1.57) 75 | 4.49 (1.79) 79 | 4.64 (1.70) 78 | 4.70 (1.83) 76 | 4.93 (1.80) 71 |
| Extraversion | 4.82 (1.05) 78 | 4.80 (1.29) 81 | 4.97 (1.17) 79 | 4.81 (1.23) 75 | 4.99 (1.04) 79 | 5.07 (1.01) 77 | 5.08 (1.07) 76 | 5.13 (1.10) 72 |

**Fig. 4 | State and trait changes across the intervention.** A–F display state (**A**, **B**), explicit trait self-concepts (**C**, **D**), and implicit trait self-concepts (**E**, **F**) for emotional stability and extraversion across time. Discontinuity markers on the y-axes indicate truncated scale ranges. Variables of (**A** and **B**) were scaled 1–7, (**C** and **D**) 1–5, and (**E** and **F**) −2 to 2. Higher values reflect higher state or trait levels. Error bars represent standard errors. Solid lines represent younger adults and dashed lines represent older adults. Sample sizes were: **A** state emotional stability, n = 76 younger/78 older participants (week 1), 73 younger/81 older (week 2), 72 younger/79 older (week 3), 72 younger/75 older (week 4), 70 younger/79 older (week 5), 66 younger/78 older (week 6), 67 younger/76 older (week 7), 63 younger/71 older (week 8); **B** state extraversion, n = 76 younger/78 older (week 1), 73 younger/81 older (week 2), 72 younger/79 older (week 3), 72 younger/75 older (week 4), 70 younger/79 older (week 5), 66 younger/77 older (week 6), 67 younger/76 older (week 7), 63 younger/72 older (week 8); **C** explicit trait emotional stability, n = 80 younger/81 older participants (T1), 74 younger/77 older (T2), 68 younger/73 older (T3); **D** explicit trait extraversion, n = 80 younger/81 older (T1), 74 younger/77 older (T2), 68 younger/73 older (T3); **E** implicit trait emotional stability, n = 73 younger/70 older participants (T1), 69 younger/72 older (T2), 63 younger/72 older (T3) ; **F** implicit trait extraversion, n = 73 younger/69 older (T1), 69 younger/72 older (T2), 63 younger/72 older (T3).

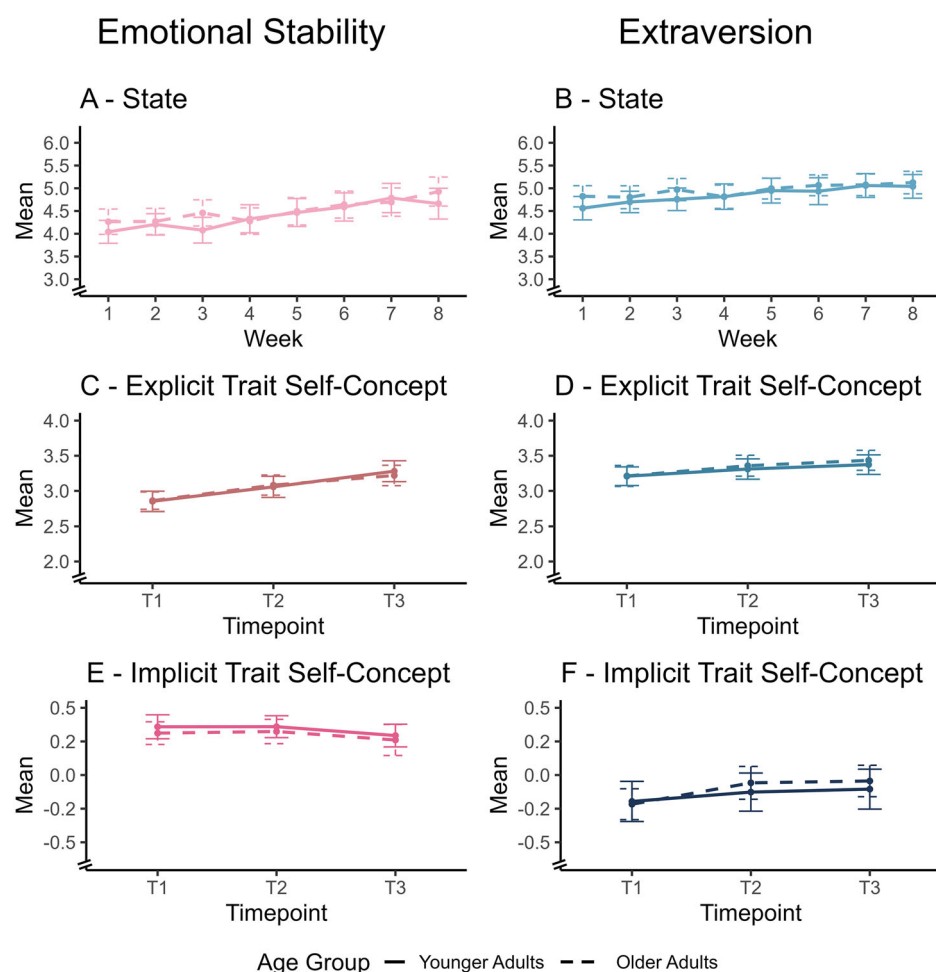

suggesting no meaningful age effects in state-trait associations across the intervention (Bayes factors = 3–20)[81].

## Exploratory analyses of long-term trait changes over 3 and 12 months

We used piecewise growth curve models to examine whether trait changes during the intervention sustained after 3 and 12 months and whether these long-term trajectories differed between younger and older adults. Figure 5 illustrates changes in standard deviations from T1 across the assessment period (see Supplementary Table S15 for all parameter estimates).

The results showed that the explicit self-concept of emotional stability remained stable after the intervention, with no credible evidence for an increase or decrease (b = −0.004, 95% CI [−0.006, 0.014]), while the explicit trait self-concept of extraversion demonstrated a small decrease (b = −0.010, 95% CI [−0.017, −0.003]). For both implicit trait self-concepts, there was no credible evidence for changes during follow-up assessments (emotional stability: b = 0.005, 95% CI [−0.002, 0.011]; extraversion: b = 0.006, 95% CI [−0.002, 0.013]). Further, there was no credible evidence for a moderating effect of age group on changes in the explicit self-concept (emotional stability: b = 0.004, 95% CI [−0.014, 0.023]; extraversion: b = −0.002, 95% CI [−0.015, 0.010]), nor the implicit self-concept of emotional stability (b = 0.010, 95% CI [−0.004, 0.022]). However, there was credible evidence for an increase in the implicit self-concept of extraversion among older adults (b = 0.016, 95% CI [0.001, 0.031]).

**Fig. 5 | Change in explicit and implicit self-concepts of emotional stability and extraversion during the personality intervention and at 3 and 12 months follow-up.** The figure displays the changes in standard deviation units for the outcome variables between the baseline (T1) and further time points. Implicit measures were not assessed at T4. The number of participants with available data at both T1 and the respective follow-up for explicit self-concepts of emotional stability and extraversion was $n = 148$ (T2), $n = 137$ (T3), $n = 121$ (T4), $n = 98$ (T5). The number of participants with available data at both T1 and the respective follow-up was for the implicit self-concepts of emotional stability $n = 124$ (T2), $n = 118$ (T3), $n = 83$ (T5), and for extraversion $n = 125$ (T2), $n = 118$ (T3), $n = 84$ (T5).

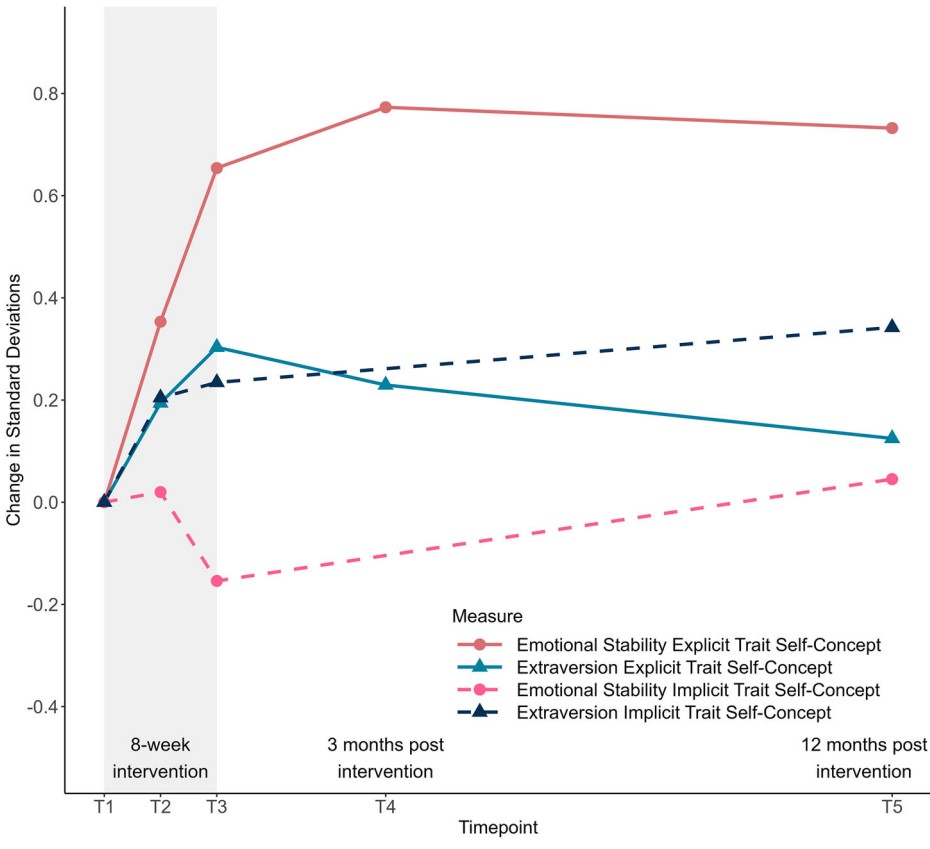

For explicit extraversion (BIC difference = 3.575) and implicit emotional stability (BIC difference = 5.032), BIC differences indicated positive evidence for the null hypothesis, suggesting no meaningful age effects in follow-up trajectories (Bayes factors = 3–20)[81]. Strong evidence for the null hypothesis was found for trait changes in the explicit self-concept of emotional stability (BIC difference = 7.033; Bayes factors = 20–150)[81]. In contrast, analyses of implicit extraversion (BIC difference = −6.159) yielded strong evidence for the alternative hypothesis, indicating age-related differences in follow-up trajectories (Bayes factors = 20–150) 80. For details, see Supplementary Table S14, Model C.

### Exploratory analyses of engagement during the intervention

Participants attended the 8-week training regularly ($M = 6.94$ sessions, $SD = 0.06$), with a moderate level of accomplished weekly tasks (scaled 1–7; $M = 4.12$, $SD = 1.17$), and engagement in practice with audio materials (scaled 1–7; $M = 3.75$, $SD = 1.56$). Yet, they had relatively few exchanges with their training buddies (scaled 1–5; $M = 2.21$, $SD = 0.71$). Participants reported that their weeks during the training were moderately hectic (scaled 1–7; $M = 3.83$, $SD = 0.97$), somewhat exhausting (scaled 1–7; $M = 3.76$, $SD = 1.00$), and moderately typical (scaled 1–7; $M = 3.91$, $SD = 1.11$).

To better understand the lack of age differences in state and trait changes, we explored age differences in these variables of engagement and daily life demands among younger and older adults employing two-sided independent-samples $t$-tests, applying Welch's $t$-tests when variances differed between groups. Although younger and older adults did not differ in their desires to improve emotional stability, $t(158.78) = 1.29$, $p = 0.199$, $d = 0.19$, 95% CI [−0.11, 0.50], younger adults had a significantly higher desire to improve their extraversion, $t(159) = 2.24$, $p = 0.026$, $d = 0.36$, 95% CI [0.05, 0.67].

Importantly, older adults reported more engagement with the intervention (see Fig. 6A): they were more engaged in weekly tasks, $t(155.15) = −4.89$, $p < 0.001$, $d = −0.77$, 95% CI [−1.08, −0.45], and audio files, $t(159.90) = −3.64$, $p < 0.001$, $d = −0.57$, 95% CI [−0.88, −0.26]. Yet,

younger and older adults reported a similar amount of contact with their training buddies, $t(157.28) = −0.95$, $p = 0.340$, $d = −0.15$, 95% CI [−0.45, 0.16], as well as practicing acquired skills in their daily lives, $t(156.19) = −1.60$, $p = 0.110$, $d = −0.26$, 95% CI [−0.57, 0.06]. Regarding context factors (Fig. 6B), younger adults reported more hectic weeks, $t(159.49) = 2.49$, $p = 0.013$, $d = 0.39$, 95% CI [0.08, 0.70], and atypical weeks, $t(158.04) = 3.23$, $p = 0.002$, $d = 0.51$, 95% CI [0.19, 0.82] than older adults during the intervention. There were no age differences in weekly exhaustion, $t(153.60) = −0.10$, $p = 0.921$, $d = −0.01$, 95% CI [−0.32, 0.30] (see Supplementary Table S16 for descriptive statistics).

### Sensitivity analyses: evaluating power to detect age-related differences

To assess the impact of the smaller sample size, we conducted a Monte Carlo simulation[77] using the observed effect sizes, comparing the actual sample size ($N = 165$) with the originally planned $N = 220$. These analyses were not pre-registered. We used a simplified latent change score model with three measurement points, one latent intercept, and one latent slope factor to achieve convergence. Although approximate, such simulations are useful for evaluating the robustness of results and the potential effects of limited statistical power[82].

The simulations showed that the observed age effects were well replicated: For both traits, estimated parameters fell within the 95% confidence interval of the population value in at least 94% of 1000 replications (see second column in Supplementary Table S17). This indicates the model reliably recovered the simulated parameters. The final column reports the proportion of statistically significant coefficients—that is, the empirical power to detect the observed effect given the sample size and model. For extraversion, age-related effects were statistically significant in only 20% of replications at $N = 165$ and 22% at $N = 220$. This indicates negligible differences in power between the intended and the realized sample sizes. While such small age effects could not be detected reliably, their magnitude is likely of little practical relevance. For emotional stability, age-related differences in

**Fig. 6 | Boxplots of age differences in engagement with the intervention and context factors.** $N = 162$. Boxplots display the distribution of engagement scores by age group. Boxes represent the inter-quartile range (IQR), with the horizontal line marking the median. Whiskers extend to data within 1.5 × IQR. Jittered points represent individual observations. Colors indicate age group (younger vs. older adults). All variables were scaled 1–7, except buddy exchange, which was scaled 1–5.

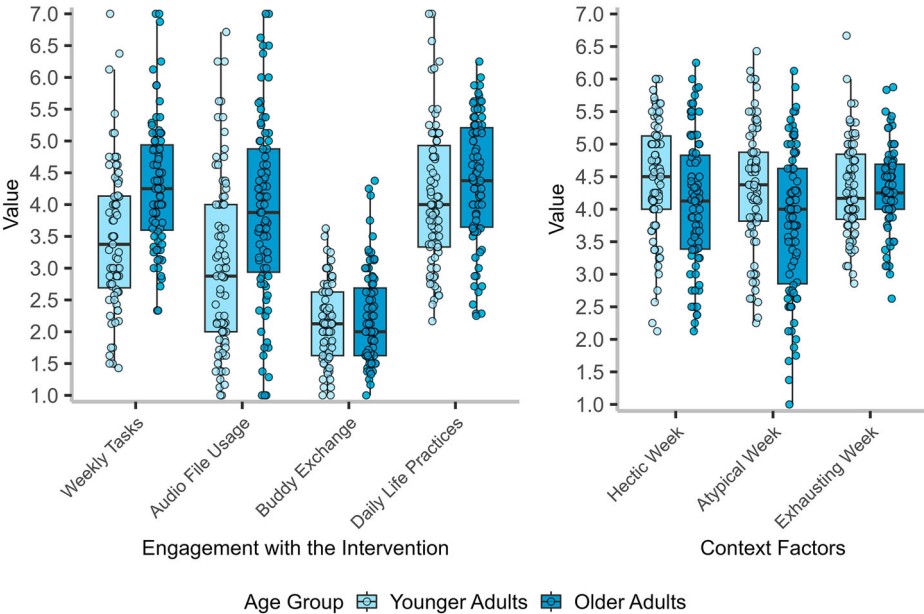

slope effects were significant in 100% of replications at both sample sizes. This stands in contrast to the empirical data, where no significant age differences were observed. The 100% detection rate likely reflects an over-estimated effect due to the narrow confidence intervals in the simulation, because the confidence intervals of the empirical results (i.e., observed data) were much wider. This difference in precision of observed and simulated effects could explain the deviation in the simulation[82]. In sum, while a larger sample might modestly increase power and estimation precision, the conclusions regarding age-related differences appear robust even with the smaller sample.

## Discussion

This longitudinal study showed that an 8-week in-person intervention focusing on socio-emotional daily processes resulted in lasting increases in personality traits of emotional stability and extraversion across younger and older adults. The findings advance theory by demonstrating (a) changes beyond trait self-reports, (b) states as underlying processes, and (c) age similarities in personality development.

Following the intervention, participants showed increases in explicit and implicit self-concepts of trait extraversion, that is, how people describe themselves in questionnaires and which characteristics people associate themselves with more indirectly[5,13], and in weekly state reports of social behavior, confirming hypotheses H1b, H2b, and H3b. Regarding emotional stability, explicit trait self-concepts and weekly states increased, but there was no consistent evidence for changes in the implicit trait self-concept, confirming H1a and H2a, but not H3a. Self-report measures are commonly criticized for being prone to demand effects and report biases, which could result in people reporting increases in personality traits after participating in interventions without increases in other manifestations of personality traits (e.g., implicit self-concepts, behavior). Thus, the current study addressed these critical shortcomings of these earlier intervention studies[14].

Consistent with H4, explicit compared to implicit self-concepts showed larger increases—implicit extraversion increased weakly and emotional stability IAT scores did not change, suggesting different magnitudes or time courses of change. Yet overlapping confidence intervals for the self-concepts of extraversion limit strong conclusions. Earlier research argues that implicit self-concepts might take more time and repetitions for implicit learning[5,25]. Despite some criticism regarding the reliability and interpretation of implicit measures such as the IAT[83], its extensive usage and insightful findings[84,85] support its complementary use for personality

research. A recent longitudinal study also observed meaningful changes in people's implicit self-concept of extraversion[27]. Still, specific reflective and associative processes, which explain individual differences in long-term changes of explicit and implicit trait self-concepts, remain poorly understood[5,41,86].

Nonetheless, the current study offers a more comprehensive under-standing of action processes underlying personality change, demonstrating that states increased during the intervention and individual differences in these increases predicted increases in explicit self-concepts of emotional stability. This is in line with earlier studies[15,27,35], and conceptual frameworks viewing state changes as precursors for trait changes[5,6,11]. Even a single, brief increase in extraverted and emotionally stable behavior can lead to (tem-poral) increases in stable trait representations of extraversion and emotional stability[41]. Yet motivation and behavior changes alone are not sufficient to elicit or explain trait changes, as emphasized in the TESSERA framework that conceptualizes behavior changes in a sequence of short-term processes—Triggering situations, Expectancy, States/State expressions, and Reactions (TESSERA5). Although behavioral changes alone account for little variance in trait changes (see also refs. [15,27,34]), reflective and associative processes are considered key in explaining why stronger trait self-concept changes may occur[5,7]. Most previous intervention studies did not examine the the-oretically proposed state-trait links and, therefore, might have missed that behavioral changes might not be sufficient to explain changes in people's explicit or implicit trait self-concepts. The current intervention addressed both reflective processes such as noticing behavioral changes and associative processes, like practicing emotion regulation and functional interpersonal behavior repeatedly during daily assignments. Yet it remains speculative, whether both pathways of personality development were addressed equally and if the extent and time courses of explicit and implicit trait changes truly differ due to different underlying processes[5,25]. In summary, intervention effects might be more lasting when not only behavioral changes occur, but self-concepts change as well[5].

As one of few studies, we examined age differences in the change processes of state and traits and found similar changes for younger and older adults in personality states, explicit and implicit traits throughout the intervention. This finding agrees with recent experimental and meta-analytic work that found no significant age differences in trait changes after experimentally induced behavioral changes[41] or psychotherapy[48,49].

These and the current findings offer further indication that smaller normative personality changes with older age[8,10] might not be fully explained

by diminished or impossible personality changes with age. Considering the results of our sensitivity analyses, we acknowledge that very small effects may not have been reliably detected with the current sample size. However, if such effects remain minimal even in the context of a strong intervention, they are probably of limited relevance in everyday contexts where opportunities for change are weaker. Instead, less pronounced normative trait changes (i.e., without intervention) might be attributed to older adults' reduced desire for change[87] and fewer life events triggering changes[9]. This reasoning extends predictions from the TESSERA framework[5], which proposes that fewer contextual changes and dampened learning processes contribute to smaller normative trait changes with older age.

The current results indicate that when older adults want to change and contextual changes occur, for example, through an (psychotherapeutic) intervention, they might compensate for potential cognitive drawbacks, such as slowing learning[39] with higher motivation. Such an interpretation is well in line with lifespan psychology that repeatedly demonstrated continuous improvement in socio-emotional functioning as people get older[88].

Interestingly, older participants matched younger participants in their desire to enhance emotional stability but reported less motivation to increase extraversion, likely reflecting prioritization of close, emotionally meaningful relationships over the formation of new social ties[89,90]. Nonetheless, increases in extraversion were equivalent across age groups, suggesting that structured behavioral engagement in a group format may help overcome baseline differences in change goals. Importantly, the self-selected sample—particularly highly motivated older adults—likely perceived trait change as feasible, which may facilitate trait changes[3]. By contrast, older adults' personality is generally viewed as less malleable[91–93]. These prevailing stereotypes may discourage engagement in interventions or reduce the likelihood of interpreting trait-incongruent states as contributing to lasting change.

Contrasting online interventions through apps and mails (see ref. 2 for a review), we deliberately carried out an in-person intervention with tasks for daily practice and a buddy-support system to enhance adherence. Notably, compared to those studies, we obtained substantially larger effect sizes regarding the explicit self-concept of emotional stability and similar effect sizes in extraversion, despite targeting extraversion only in the latter four weeks (emotional stability average $d = 0.33$, extraversion average $d = 0.38$[2]). We speculate that we would have seen even larger changes with durations similar to the ones of digital interventions (i.e., 12–16 weeks) because the in-person format allowed training interpersonal behavior directly and direct trainer feedback likely reinforced outcomes, whereas digital interventions rely exclusively on the participant's adherence to the exercises in their daily lives. Nevertheless, the effectiveness of digital interventions is still impressive and allows the inclusion of more people simultaneously than in-person interventions[2].

## Limitations

General limitations of the study include that it remains unclear, which processes and aspects are most relevant for altering personality traits in the direction that people desire[2,33]. Future intervention studies could compare different treatments to gauge the differential contributions of intervention parts (e.g., ref. 32). At the same time, as in psychotherapy and mental well-being interventions, such differences in treatment conditions might not be large[48,94]. Potential reasons are (a) common change factors that are included to most interventions and most powerful, while other factors play only a minor role[16] and (b) limiting people to specific aspects of the treatment (e.g., social feedback, reflection) might be impossible in psychological treatments compared to pharmacological intervention, where provided substances can be strictly controlled (see ref. 41 for similar arguments). Second, we measured states and daily life processes only in weekly assessments to reduce participant burden. More fine-grained assessments of behavioral, cognitive, and emotional changes in people's daily lives would enable an even better understanding of the temporal dynamics of personality changes. Ultimately, overlapping with urgent questions of psychotherapy, the understanding of the daily life processes would greatly facilitate explaining individual differences in the strength and sustainability of behavioral and self-concept changes[48,94].

In sum, this multimethod study offers a detailed view of how younger and older adults' socio-emotional personality traits can change through an 8-week in-person intervention. Across the intervention, state extraversion and emotional stability increased, which partly explained the increases in trait extraversion and emotional stability. The changes were largely robust over the following 12 months. These findings underscore the potential of structured socio-emotional training to promote lifelong psychological adaptability in aging societies.

## Data availability

The anonymized raw and processed data supporting the findings of this study are publicly available on the Open Science Framework (OSF). They can be accessed in the folder "Files/Data and Codebook" at https://doi.org/10.17605/OSF.IO/AC4B2. The dataset includes scores from demographic questions, questionnaire items, and D2-scores from the Implicit Association Test.

## Code availability

The code to replicate the findings can be accessed in the folder "Files/Code and Mplus Output" from https://doi.org/10.17605/OSF.IO/AC4B2.

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

## Acknowledgements
This research has been funded by the German Research Foundation (Deutsche Forschungsgemeinschaft; DFG), project number 260006982, with grants to Cornelia Wrzus (WR 160/1-2) and Corina Aguilar-Raab (AG 250/3-2). The funders had no role in study design, data collection and analysis, decision to publish, or preparation of the manuscript. The authors gratefully acknowledge the research assistants and students who supported the data collection of this study. Special thanks are extended to Alina Pees, Guy Pires Cabrita, Andreas Spiziali, Mattis Nuding, and Marie Noack for their dedicated assistance. The authors also thank Oliver Schilling for his valuable statistical advice. For the publication fee, we thankfully acknowledge financial support by Heidelberg University.

## Author contributions
G.K. contributed through conceptualization, investigation, methodology, project administration, data curation, formal analysis, visualization, writing the original draft, writing–review & editing. K.S.A.B. contributed to conceptualization, investigation, methodology, project administration, data curation, writing–review & editing. C.A.R. contributed to conceptualization, funding acquisition, methodology, project administration, writing–review & editing. W.B. contributed through writing–review & editing. J.W. contributed through writing–review & editing. C.W. contributed through conceptualization, funding acquisition, methodology, project administration, resources, supervision, writing the original draft, writing–review & editing.

## Funding

## Competing interests
The authors declare no competing interests.
