## [Transparent Peer Review file · Communications Psychology]

Personality Intervention Affects Emotional Stability and Extraversion Similarly in Older and Younger Adults

Corresponding Author: Dr Cornelia Wrzus

Version 0:

Decision Letter:

Dear Dr Wrzus,

Thank you for your patience during the peer-review process. Your manuscript titled "Similar Personality Changes among Younger and Older Adults: Findings from a Multi-Method Intervention Study on Socio-Emotional States and Traits" has now been seen by 2 reviewers, and I include their comments at the end of this message. They find your work of interest but raised some important points. We are interested in the possibility of publishing your study in Communications Psychology, but would like to consider your responses to these concerns and assess a revised manuscript before we make a final decision on publication.

We therefore invite you to revise and resubmit your manuscript, along with a point-by-point response to the reviewers. Please highlight all changes in the manuscript text file.

Editorially, we consider it important that the revision addresses Reviewer 2's concerns regarding the modeling and model fit as well as R1's concerns regarding power by providing a sensitivity power analysis.

I am attaching an Editorial Requests Table that details critical reporting requirements for the revised manuscript. Please attend to each item and ensure your manuscript is fully compliant. If your revised manuscript is not aligned with these requests on major issues, such as those concerning statistics, it may be returned to you for further revisions without re-review.

Please submit the following items:

- Revised manuscript
- Point-by-point response to the referees' comments
- Cover letter (as a separate document)
- <https://www.nature.com/documents/nr-reporting-summary.zip>>Nature Research Reporting Summary
- <https://www.nature.com/documents/nr-editorial-policy-checklist.pdf>>Editorial Policy Checklist
- Completed Editorial Request Table (attached).

via this link: Link Redacted .

Additional guidance is available in our style and formatting guide Communications Psychology formatting guide.

Best regards,

Jennifer Bellingtier

on behalf of
Yana Fandakova
External Editor
Communications Psychology

REVIEWER EXPERTISE:

Reviewer #1 lifespan development, longitudinal studies
Reviewer #2 lifespan development, longitudinal studies

REVIEWER REPORTS:

Reviewer #1 (Remarks to the Author):

In the current (preregistered) study, the authors investigate the effects of an intervention to change neuroticism and extraversion in younger and older adults. N = 168 participants (aged 18-35 and 55+) underwent an 8-week group intervention focusing on stress, emotion regulation, and socio-emotional competences. Personality was assessed several times during and after the intervention, measuring both implicit and explicit measures of both states and traits. The intervention was effective in changing explicit and implicit traits, as well as states of extraversion, and explicit traits and states, but not implicit traits of neuroticism. Changes in states were only partially related to changes in traits. Interestingly, there were no age differences in the intervention effects, but older adults reported more engagement with the intervention.

I am very impressed by this paper. The topic of the study is novel and address important current topics in research on personality development: The impact and mechanisms of personality change interventions and whether they function similarly across the lifespan. The intervention, methodology and analyses are very detailed and sophisticated, the results very relevant and the paper is also very well written. The documentation of the analyses and results is extensive. The conclusions drawn are justified by the empirical findings. Limitations are clearly addressed.

In only have a couple of minor issues and thoughts the authors could address in a revision:

- In the abstract, the authors could mention that they address only Extraversion and Emotional Stability and not all Big Five traits.
- The association of personality states and traits could be introduced in a bit more detail in the introduction, as many readers of the journal might not be familiar with dual-process models of personality. This might also give more context to understand the findings that state and trait change are not systematically associated for all variables.
- The final sample was quite a bit lower compared to the sample size indicated by the power analysis. Was there any systematic reason for the substantial dropout and what are the consequences regarding statistical power of the study?
- Please report the age range for the older participant group in addition to the mean age.
- What was the reason for the enrolment fee?
- Were all assessments (also those at T4 and T5) done on site or online?
- If I understood correctly, the groups were mixed (that is included both younger and older adults)?
- All the trainers were young adults, might this have affected the older adults' perception of the intervention and their engagement?
- The summary at the beginning of the results section should be integrated in the "current study" part.

- In terms of age differences, one important aspect why older adults might show less trait or personality change in general (or report less motivation to change) is the prevailing stereotype that older adults are set in their personality and that personality change is not possible in older adults. This might keep people from engaging in non-trait consistent behavior or from signing up for personality change interventions. It is interesting to see that the older adults in this study reported more commitment to the study. Nevertheless, I think personality change is something that for many older adults is not something they would think they can achieve. This could be mentioned in the discussion section.

- In terms of the desire to change certain traits, which was lower for extraversion in the older group, I was also wondering whether the authors considered motivational changes in late life, as for instance described in socioemotional selectivity theory (Carstensen, 1995). According to these theories, older adults should be more motivated by social goals (e.g., increase in agreeableness) whereas for younger adults, knowledge-related goals should be more important (e.g., increase in extraversion). This could have an impact on which traits people might want to change, and could be an interesting avenue for future research. Relatedly, as the intervention was a group intervention, this might be more beneficial for older adults' intervention effects. Again, this is just a thought I had that could be integrated into the discussion.

Reviewer #2 (Remarks to the Author):

In the manuscript "Similar Personality Changes among Younger and Older Adults: Findings from a Multi-Method Intervention Study on Socio-Emotional States and Traits", the authors investigate the effects on personality from an 8-week intervention program.

I think the study provides a significant contribution to the literature. It is overall well-written and the study has an interesting design that – as far as I can tell – has been conducted in the best possible way. (I am not a fan of implicit measures; as I don't know what they are actually measuring; but I can see that including them here provides a nice counterpoint.)

There are a few points that might need clarification:

1) I had some difficulties initially understanding what the state part is. For a few seconds, I thought you may look at variability/fluctuations over time. It took me longer than I want to admit until I understood that it was about the weekly assessments. I think what tripped me off was naming it "state" vs. "trait". The difference between the weekly assessment and the roughly 4-week intervals between pre-, middle-, and post-assessment don't seem that big to justify the state vs. trait differentiation. For someone, who is doing experience-sampling or daily diary work, weeks are like traits. For someone looking at age-related change in adulthood, even one year might be "state" in terms of how small the change is.

I think the authors might want to clarify this a bit in the wording and how they talk about it. Maybe instead of "state" one could use "weekly"?! Maybe some sentences here and there could clarify that.

But it also brings to mind that given the similarity in the time frame, it seems less surprising that the "state" and "trait" factors are related. Though, still a nice model.

2) The beginning of the result section seems to intermix some repetitive information from the method section and new details that might be better located in a procedure section. Maybe cleaning that section up a bit and remodeling some info into the method section.

3) I think the biggest problems are associated with the models, clarity around the models, and their model fit. First, are the model fits presented in Table S5 for the Models A? I think that needs to be clarified. It also needs dfs for the models. And it is somewhat strange to report results before reporting the model fits. For the complexity of these models, the model fits look rather great. Given how these models look, I would have expected worse fit or convergence issues. And I am saying that because it seems Models B & C did not converge, which is a problem. Second, where are the values for IS2 and IS3? I understand that they are method factors, but they should be reported somewhere. And are they really necessary? These models often work well without including a method factor for the items. And see next point!

Third, I frequently encounter convergence issues with these type of models with extra method factors. There is a reason why correlated uniqueness models are often recommended as they often provide more stable model pattern. But it seems another simpler method would be to treat the trait variables as manifest variables (e.g. average of p1, p2, and p3) rather than latent variables. That would immediately simplify the model but also create more robust models – at least from my experience. And it would remove the necessity for IS2 and IS3.

Fourth, why is age only predicting the change or slope in these models (and not the intercept)? It seems table S2 suggest that there are no age differences in the state variables. Do we also have that for the traits? I may have missed it but it probably deserves a paragraph in the results to talk about age differences (in means).

Finally, I found the model sections were leaning on the technical side in the writing style. I think a few more sentences would be helpful to clarify what the purpose of these different models is. I think the reader is easily lost here.

4) Shouldn't there be a table with the descriptives (means, SDs) for the trait and state variables somewhere? And by age group? Like S15 has the correlations, but no means (SDs).

Minor issues:

- define ES and EX & include dfs if chi-square values are reported (Table S5, S8, S10, S16)
- Figure 4 looks as if it was resized unevenly; at least the axes font looks somewhat strange. Maybe redo the font and maybe make the legend slightly larger. I would also suggest to include different markers/lines (e.g. dashed) for the four groups to make it easier for people with vision problems to differentiate the lines.
- In Table S11, isn't it obvious that there is a difference. Does it really need subscripts?

Version 1:

Decision Letter:

Dear Dr Wrzus,

Your manuscript titled "Similar Personality Changes among Younger and Older Adults: Findings from a Multi-Method Intervention Study on Socio-Emotional States and Traits" has now been seen by our reviewers, whose comments appear below. In light of their advice I am delighted to say that we are happy, in principle, to publish a suitably revised version in Communications Psychology.

We therefore invite you to revise your paper one last time to address the remaining concerns of our reviewers and a list of editorial requests. At the same time we ask that you edit your manuscript to comply with our format requirements and to maximise the accessibility and therefore the impact of your work.

EDITORIAL REQUESTS:

SUBMISSION INFORMATION:

OPEN ACCESS:

Communications Psychology is a fully open access journal. Articles are made freely accessible on publication. For further information about article processing charges, open access funding, and advice and support from Nature Research, please

visit <https://www.nature.com/commpsychol/open-access>

* **DATA AVAILABILITY:**

Link Redacted

Best regards,

Jennifer Bellingtier

Jennifer Bellingtier, PhD
Senior Editor
Communications Psychology

on behalf of
Yana Fandakova
External Editor
Communications Psychology

REVIEWER EXPERTISE:

Reviewer #1 lifespan development, longitudinal studies

REVIEWERS' COMMENTS:

Reviewer #1 (Remarks to the Author):

I appreciate the authors' thorough revision of the manuscript and their responsiveness regarding the issues raised. I think the paper will make an important contribution to the literature.

** Visit Nature Research's author and referees' website at <http://www.nature.com/authors> for information about policies, services and author

benefits**

AUTHOR RESPONSE: Personality State-Trait Associations as Developmental Processes in Younger and Older Adults

We hereby resubmit our revised manuscript 'Similar Personality Changes among Younger and Older Adults: Findings from a Multi-Method Intervention Study on Socio-Emotional States and Traits'.

We sincerely appreciate the reviewers' recognition of the quality of our work and the constructive feedback provided to further improve our manuscript. In response to the feedback, we have carefully considered and addressed each of the reviewers' comments.

All revisions made to the manuscript have been highlighted in blue font. Our detailed point-by-point responses to each comment are presented below.

In addition, we have rephrased and condensed several paragraphs to reduce the word count and made further modifications to comply with the formatting guidelines. For example, we converted the reference format, Figure 4 from bar plots to box plots, removed bold and cursive text from tables, and added separate sections for data and code availability. These changes are also highlighted in blue.

Reviewer #1 (Remarks to the Author):

In the current (preregistered) study, the authors investigate the effects of an intervention to change neuroticism and extraversion in younger and older adults. N = 168 participants (aged 18-35 and 55+) underwent an 8-week group intervention focusing on stress, emotion regulation, and socio-emotional competences. Personality was assessed several times during and after the intervention, measuring both implicit and explicit measures of both states and traits. The intervention was effective in changing explicit and implicit traits, as well as states of extraversion, and explicit traits and states, but not implicit traits of neuroticism. Changes in states were only partially related to changes in traits. Interestingly, there were no age differences in the intervention effects, but older adults reported more engagement with the intervention.

I am very impressed by this paper. The topic of the study is novel and address important current topics in research on personality development: The impact and mechanisms of personality change interventions and whether they function similarly across the lifespan. The intervention, methodology and analyses are very detailed and sophisticated, the results very relevant and the paper is also very well written. The documentation of the analyses and results is extensive. The conclusions drawn are justified by the empirical findings. Limitations are clearly addressed.

In only have a couple of minor issues and thoughts the authors could address in a revision:

#R1_1: In the abstract, the authors could mention that they address only Extraversion and Emotional Stability and not all Big Five traits.

RESPONSE: We appreciate the reviewer's feedback and included the suggestion in the abstract (p. 2):

"In this preregistered, multi-method study, we examined associations between changes in personality states and explicit and implicit trait self-concepts of emotional stability and extraversion after an 8-week socio-emotional intervention, 3 and 12 months later."

#R1_2: The association of personality states and traits could be introduced in a bit more detail in the

introduction, as many readers of the journal might not be familiar with dual-process models of personality. This might also give more context to understand the findings that state and trait change are not systematically associated for all variables.

RESPONSE: We appreciate your valuable suggestion to provide more context regarding the association of personality states and traits within the dual-process model. In response, we have elaborated on this topic in greater detail in the introduction (p. 4):

"Personality states represent momentary feelings, thoughts, and behaviors related to trait-domains (Fleeson & Gallagher, 2009) and are considered situation-dependent manifestations of personality traits (Back et al., 2009; Baumert et al., 2017; Hampson, 2012). At the same time, trait-congruent states presumably reinforce stability, while incongruent states promote trait changes under certain conditions (Wrzus & Roberts, 2017). States are often assessed momentarily via experience sampling (e.g., (Wrzus et al., 2021; Wrzus & Roberts, 2017)) or over short periods of time, such as the past day or week to capture short-term expressions of traits (Allemand et al., 2024; Borghuis et al., 2019). State emotional stability reflects reactions to difficult situations and emotional intensity and stability (Mader et al., 2023; Suls & Martin, 2005; Wrzus et al., 2021). State extraversion involves social and assertive behavior, positive affect, and high energy in social situations (Smillie et al., 2015; Van Zalk et al., 2020). Consistent with dual-processes models of personality, we conceptualize personality traits being mentally represented in two systems, explicit and implicit self-concepts (Back et al., 2009; Rauthmann, 2024; Wrzus & Roberts, 2017). The explicit self-concept represents consciously accessible self-evaluations, typically measured via questionnaires. In contrast, the implicit self-concept comprises rather automatic associations between the self and personality characteristics, accessible through indirect measures such as the Implicit Association Test (IAT)²³. Previous research found moderate links between explicit and implicit self-concepts (Back et al., 2009; Hofmann et al., 2005), likely due to different accessibility, methodological factors, and also developmental pathways (Baumert et al., 2017; Gawronski & Bodenhausen, 2006; Wrzus & Roberts, 2017). States contribute to changes in explicit and implicit traits through different pathways: Changes in automatic components shape implicit self-concepts through associative processes like reinforcement learning (Gawronski & LeBel, 2008; Wrzus & Roberts, 2017). For example, repeated positive feedback for staying calm under pressure may strengthen the implicit "me-calm" association tied to emotional stability. In addition, deliberate aspects of states correspond to the explicit self-concept and presumably change through reflective processes such as self-reflection. Reflection can (a) incorporate repeated experiences into the explicit self-concept (e.g., interpreting calmness as increased stability), and (b) support self-regulation by bridging gaps between actual and desired self via novel states (e.g., applying

stress management techniques; (Back et al., 2009; Baumert et al., 2017; Gawronski & LeBel, 2008; Wrzus & Roberts, 2017)). "

#R1_3: The final sample was quite a bit lower compared to the sample size indicated by the power analysis. Was there any systematic reason for the substantial dropout and what are the consequences regarding statistical power of the study?

RESPONSE: The dropout rate in our study was 18%, which is considerably lower than in comparable online intervention studies (e.g., 37% in Stieger et al., 2021) and similar to previous in-person intervention study (Allan et al., 2018). Approximately half of the dropouts never initiated the training. In most cases, the reasons for nonparticipation are unknown, but some participants reported scheduling conflicts, such as extended vacations or increased work commitments that prevented engagement.

Among those who started but did not complete the training (approximately 9%), several reported unforeseen negative life events (e.g., bereavement, illness) or other personal issues that made continued participation impossible.

A more substantial challenge was recruiting a sufficient number of eligible participants. Of the 1,019 individuals who consented to screening, only 45% met the inclusion criteria. Specifically, 18.8% were excluded because they were currently in psychotherapy, engaged in another training program, or lacked internet access or suitable hardware. An additional 26% were excluded due to indications of mental health concerns (e.g., depression, anxiety, suicidal ideation) or incomplete screening questionnaires (see Figure S1 for an overview). We had to exclude several people from participation despite their interest, for the sake of the participants: Although the graduate student trainers went through a thorough train-the-trainer formation, had extensive experience in counseling, and were supervised by two experienced professionals, they were not licensed psychotherapists. Combined with the setting of a non-clinical, once a week group intervention, participants with more severe mental health issues would not have received suitable and appropriate support for their challenges.

Hence, the overall exclusion rate of 45% was higher than in other studies (e.g., 32% in Stieger et al., 2021). One important distinction is that our intervention targeted socio-emotional traits closely related to mental health, whereas Stieger et al. focused on all Big Five traits. This narrower focus may have especially attracted individuals with existing mental health issues.

We revised the corresponding paragraph to provide more information about the sample recruitment, exclusions, and dropout in the main manuscript (p.13):

"In total, 1,019 individuals completed the screening. Of these, 45% ($n = 458$) were excluded based on predefined criteria, such as elevated scores on the mental health screening or ongoing psychotherapy/other mental-health trainings. Of those eligible, 63.7% ($n = 357$, 75% female, 25 % male, 59 % younger age group) did not enroll in the training. Overall, 203 individuals (19.9% of screened people) enrolled, of which 20 dropped out before the training started, and 18 did not complete the training (i.e., ≤ 4 training sessions). Enrollment occurred in three cohorts in January, April, and June 2023. Drop-out analyses showed no substantial

differences between completers and dropouts (Supplementary Table S3). Supplementary

Figure S1 details the study and attrition flow."

We had originally planned two training cohorts and extended the project to a third cohort to increase the sample size. Further extensions were not feasible due to time constraints and trainer availability.

To assess the impact of the smaller sample size, we conducted a Monte Carlo simulation using the observed effect sizes, comparing the actual sample size ($N = 165$) with the originally planned $N = 220$. We had to use a simplified latent change score model for simulations with three measurement points, one latent intercept, and one latent slope factor, to achieve convergence. While being only an approximation, such simulations are useful for evaluating the robustness of result and the potential effects of limited statistical power (Maxwell et al., 2008).

The simulations showed that the observed age effects were well replicated: For both traits, estimated parameters fell within the 95% confidence interval of the population value in at least 94% of 1,000 replications (see second column in the table). This indicates the model reliably recovered the simulated parameters. The final column reports the proportion of statistically significant coefficients—that is, the empirical power to detect the observed effect given the sample size and model. For extraversion, age-related differences in changes over time were statistically significant in only 20% of replications at $N = 165$ and 22% at $N = 220$. This indicates negligible differences in power between the intended and the realized sample sizes. While such small age effects could not be detected reliably, their magnitude is likely of little practical relevance.

For emotional stability, age-related differences in slope effects were significant in 100% of replications at both sample sizes. This stands in contrast to the empirical data, where no significant age differences were observed. The 100% detection rate likely reflects an overestimated effect due to the narrow confidence intervals in the simulation, because the confidence intervals of the empirical results (i.e., observed data) were much wider. This difference in precision of observed and simulated effects could explain the unexpected finding in the simulation (Maxwell et al., 2008). In sum, while a larger sample might modestly increase power and estimation precision, the conclusions regarding age-related differences appear robust even with the smaller sample.

We now report these analyses also in the main manuscript (pp. 26-27) with a table providing an overview of the relevant parameters for the age effects on intercepts and slopes in the Supplementary material Table S17 (p. 20)

Table S17

Results of Monte Carlo Simulations Comparing the Planned and Final Sample Sizes

Parameters	% models with estimate within 95% CI	% models with sign. estimate
Extraversion, $N = 165$		
age effects in intercept $b = -.01$	96%	4%
age effects in slope $b = .01$	94%	20%
Extraversion, $N = 220$		
age effects in intercept $b = -.01$	95%	5%
age effects in slope $b = .01$	94%	22%
Emotional stability, $N = 165$		
age effects in intercept $b = .02$	95%	6%
age effects in slope $b = .07$	95%	100%
Emotional stability, $N = 220$		

age effects in intercept $b = .02$	95%	6%
age effects in slope $b = .07$	95%	100%

Note. We always used 1000 replications.

Further, we included the findings in the discussion (p. 29)

"Considering the results of our sensitivity analyses, we acknowledge that very small effects may not have been reliably detected with the current sample size. In the context of the strong intervention-based induction of state changes and reflective or associative processes, such very small effects—if present in the general population—are probably of little practical relevance."

#R1_4: Please report the age range for the older participant group in addition to the mean age.

RESPONSE: We now included the age range in the sample description (p.13)

"The final sample ($N = 165$) that participated in the intervention consisted of 80 younger adults ($M_{Age} = 28.33$, $SD_{Age} = 4.92$, $range_{Age} = 19-42$, 75 % female, 24 % male, 1 % non-binary, 65 % university degree) and 85 older adults ($M_{Age} = 63.55$, $SD_{Age} = 7.20$, $range_{Age} = 50-78$, 75 % female, 25 % male, 60% university degree). The ages of six individuals fell between the predefined criteria for younger adults (18–35 years) and older adults (50+ years). They were categorized as younger adults, as their ages were closer to the mean and range of this group."

#R1_5: What was the reason for the enrolment fee?

RESPONSE: We implemented an enrollment fee as previous research showed that it significantly increases the study adherence and compliance (see e.g., Van Dieren et al., 2013; Wesch et al., 1987). For clarification we included this reasoning in the manuscript (p. 13):

"Based on evidence that enrollment fees improve adherence (e.g., Van Dieren et al., 2013; Wesch et al., 1987), we charged a fee of 80 EUR (50 EUR reduced fee for individuals with low income, e.g., students/pensioners). Participants received a reimbursement of up to 110 EUR (completing all questionnaires) and a 50% fee refund when attending ≥ 5 sessions."

#R1_6: Were all assessments (also those at T4 and T5) done on site or online?

RESPONSE: All assessments, including those at T4 and T5, were conducted online. We have now added this information on p. 13:

"All assessments were conducted online using the SosciSurvey platform (Leiner, 2022)."

#R1_7: If I understood correctly, the groups were mixed (that is, included both younger and older adults)?

RESPONSE: Yes, they were mixed due to organizational constraints and the assumption that both age groups could benefit from each other's perspectives. We now clarified this on p. 10:

"Groups included both younger and older adults."

#R1_8: All the trainers were young adults—might this have affected the older adults' perception of the intervention and their engagement?

RESPONSE: Yes, the trainers were exclusively younger adults. Participants were informed that their trainers were graduate students completing a Master's-level course. We did not expect that older participants would perceive the training content as primarily relevant to younger adults.

Nevertheless, we understand your concern regarding the potential impact of age differences on participant engagement and alliance. Prior clinical research has demonstrated that the initial affiliation between therapists and clients is not negatively affected by age differences, and that the development of affiliation over time may even be stronger when therapists are younger than their clients (Behn et al., 2018). Furthermore, a meta-analysis indicated that the overall effects of the therapist-client alliance on outcomes are small in group therapy settings (Alldredge et al., 2021). Therefore, it is plausible that age differences between trainers and participants were of limited relevance in our group-based intervention context.

We have now added in the main text of the manuscript (pp. 11-12):

"All trainers were younger adults. Previous research suggests that age discrepancies between therapist and client do not weaken the therapeutic alliance and may even strengthen it in the case of younger therapists (Behn et al., 2018). Further, overall effects of the therapist-client alliance on outcomes are small in group therapy settings (Alldredge et al., 2021)."

#R1_9: The summary at the beginning of the results section should be integrated in the "current study" part.

RESPONSE: We agree and moved the information partly into the present research (pp.8-9), partly into the Method section (p. 11).

#R1_10: In terms of age differences, one important aspect why older adults might show less trait or personality change in general (or report less motivation to change) is the prevailing stereotype that older adults are set in their personality and that personality change is not possible in older adults. This might keep people from engaging in non-trait-consistent behavior or from signing up for personality change interventions. It is interesting to see that the older adults in this study reported more commitment to the study. Nevertheless, I think personality change is something that, for many older adults, is not something they would think they can achieve. This could be mentioned in the discussion section.

RESPONSE: We agree that age stereotypes and perceived feasibility of change may be relevant factors that differ in our sample compared to the general population. Therefore, we have incorporated this suggestion into the discussion (p. 30):

"Importantly, the self-selected sample—particularly highly motivated older adults—likely perceived trait change as feasible, which may facilitate trait changes (Lücke et al., 2020). By

contrast, older adults' personality is generally viewed as less malleable (Chaney & Chasteen, 2024; Finkelstein et al., 1995; Wrenn & Maurer, 2004). These prevailing stereotypes may discourage engagement in interventions or reduce the likelihood of interpreting trait-incongruent states as contributing to lasting change."

#R1_11: In terms of the desire to change certain traits, which was lower for extraversion in the older group, I was also wondering whether the authors considered motivational changes in late life, as for instance described in socioemotional selectivity theory (Carstensen, 1995). According to these theories, older adults should be more motivated by social goals (e.g., increase in agreeableness), whereas for younger adults, knowledge-related goals should be more important (e.g., increase in extraversion). This could have an impact on which traits people might want to change, and could be an interesting avenue for future research. Relatedly, as the intervention was a group intervention, this might be more beneficial for older adults' intervention effects. Again, this is just a thought I had that could be integrated into the discussion.

RESPONSE: We appreciate the reviewer's suggestion and therefore included a discussion of our findings with those of socioemotional selectivity theory (p. 31):

"Interestingly, older participants matched younger participants in their desire to enhance emotional stability but reported less motivation to increase extraversion, likely reflecting prioritization of close, emotionally meaningful relationships over the formation of new social ties (Carstensen, 1995; Lang & Carstensen, 1994). Nonetheless, increases in extraversion were equivalent across age groups, suggesting that structured behavioral engagement in a group format may help overcome baseline differences in change goals. "

Reviewer #2 (Remarks to the Author):

In the manuscript "Similar Personality Changes among Younger and Older Adults: Findings from a Multi-Method Intervention Study on Socio-Emotional States and Traits", the authors investigate the effects on personality from an 8-week intervention program.

I think the study provides a significant contribution to the literature. It is overall well-written and the study has an interesting design that – as far as I can tell – has been conducted in the best possible way. (I am not a fan of implicit measures; as I don't know what they are actually measuring); but I can see that including them here provides a nice counterpoint.)

There are a few points that might need clarification:

#R2_1: I had some difficulties initially understanding what the state part is. For a few seconds, I thought you may look at variability/fluctuations over time. It took me longer than I want to admit until I understood that it was about the weekly assessments. I think what tripped me off was naming it "state" vs. "trait". The difference between the weekly assessment and the roughly 4-week intervals between pre-, middle-, and post-assessment don't seem that big to justify the state vs. trait differentiation. For someone doing experience-sampling or daily diary work, weeks are like traits. For someone looking at age-related change in adulthood, even one year might be "state" in terms of how small the change is. I think the authors might want to clarify this a bit in the wording and how they

talk about it. Maybe instead of “state” one could use “weekly”? Maybe some sentences here and there could clarify that. But it also brings to mind that given the similarity in the time frame, it seems less surprising that the “state” and “trait” factors are related. Though, still a nice model.

RESPONSE: We appreciate the reviewer’s thoughtful comment regarding the use of the term “state”. The term “state” is somewhat diverse across papers, may be assessed referring to specific situations (Quintus et al., 2021), as a daily aggregate (Borghuis et al., 2020), or weekly aggregates (Tamm et al., 2024). We describe in the following why we consider it to be in line with our conceptualization.

We chose the term “state” for two main reasons:

(1) Theoretical rationale and matching assessment:

The conceptual distinction between states and traits follows theoretical work suggesting that changes in relatively stable self-concepts (traits) are preceded by short-term behavioral and experiential expressions of traits, that is, states (Fleeson & Gallagher, 2009). Although our weekly assessments do not capture momentary expressions, they reflect proximal behaviors rather than global self-concepts, because participants rated their weekly behavior on a scale from “stressed” to “relaxed” (state). In contrast, traits refer to more stable self-concepts and were assessed as general evaluations without a specific time frame, e.g., “I am someone who is relaxed and handles stress well.” Typically, these trait measures are used in yearly assessment and demonstrating substantial changes over shorter time periods also speaks to the malleability of traits (e.g., also in psychotherapy, Roberts et al., 2017). In summary, although temporally aggregated on a weekly basis, our weekly measures target specific behaviors, not broader personality self-concepts, whereas our trait measures target general self-concepts and were assessed more frequently than in observational studies of personality development (e.g., Bleidorn et al., 2022).

(2) Terminological consistency:

We aimed to align with existing terminology in personality development research. Introducing a new label (e.g., “weekly ratings”) may reduce comparability and clarity. However, we agree that our use of “state” could be better defined early in the manuscript.

We agree with the reviewer that it should be clarified earlier in the manuscript how we defined and assesses personality traits. We have therefore included in the introduction (p.4):

“Personality states represent momentary feelings, thoughts, and behaviors related to trait-domains (Fleeson & Gallagher, 2009) and are considered situation-dependent manifestations of personality traits (Back et al., 2009; Baumert et al., 2017; Hampson, 2012). (...) States are often assessed momentarily via experience sampling (e.g., (Wrzus et al., 2021; Wrzus & Roberts, 2017)) or over short periods of time, such as the past day or week to capture short-term expressions of traits (Allemand et al., 2024; Borghuis et al., 2019) ”

And in more detail in the present research (p. 7):

“While the term “state” is commonly used in experience sampling studies to describe situation-specific experiences (Quintus et al., 2021)), here we refer to week-to-week changes in states, capturing short-term expressions compared to general trait assessments. Although weekly averages of states are less temporally fine-grained than momentary assessments,

they remain distinct from trait self-concepts, as they reflect recent behavioral patterns rather than generalized self-evaluations. For example, participants rate how they behaved during the past week (e.g., "stressed" vs. "relaxed"). In contrast, traits refer to more stable self-concepts, which are assessed as general evaluations without a specific time frame (e.g., "I am someone who is relaxed and handles stress well"). Thus, weekly assessments remain consistent with theoretical conceptualizations of personality states as behaviorally anchored short-term constructs."

We further clarified in the description of measures that trait measures were general self-evaluations and state measures referred to the past week (pp.14-15):

"Explicit personality self-concepts were assessed as general evaluations without a specific time frame using the Big Five Inventory-2 (Danner et al., o. J.; Soto & John, 2017) (e.g., "I am someone who is relaxed and handles stress well."). "

"Participants reported their personality states during the past week using six bipolar items for emotional stability (e.g., *stressed* versus *relaxed*) and four items for extraversion (e.g., *shy* versus *talkative*). "

Also, we addressed this matter within the limitations of the manuscript (p.31):

"Second, we measured states and daily life processes only in weekly assessments to reduce participant burden. More fine-grained assessments of behavioral, cognitive, and emotional changes in people's daily lives would enable an even better understanding of the temporal dynamics of personality changes. "

#R2_2: The beginning of the result section seems to intermix some repetitive information from the method section and new details that might be better located in a procedure section. Maybe clean that section up a bit and move some info to the method section.

RESPONSE: We agree and moved the information partly into the present research (pp.8-9), and partly into the Method section (p. 11). This response is copy pasted from R1_9.

#R2_3: I think the biggest problems are associated with the models, clarity around the models, and their model fit. First, are the model fits presented in Table S5 for the Models A? That needs to be clarified. It also needs dfs for the models. And it is somewhat strange to report results before reporting the model fits. For the complexity of these models, the model fits look rather great. Given how these models look, I would have expected worse fit or convergence issues. And I am saying that because it seems Models B & C did not converge, which is a problem. Where are the values for IS2 and IS3? I understand they are method factors, but they should be reported somewhere. And are

they really necessary? These models often work well without including a method factor for the items. Correlated uniqueness models are often recommended as they provide more stable model patterns. A simpler method would be to treat the trait variables as manifest variables (e.g., average of p1, p2, and p3) rather than latent variables. That would simplify the model and create more robust models – and remove the need for IS2 and IS3. Why is age only predicting the change or slope in these models and not the intercept? Table S2 suggests there are no age differences in the state variables. Do we also have that for the traits? This probably deserves a paragraph in the results to talk about age differences (in means). I found the model sections leaned on the technical side in the writing style. A few more sentences would help clarify what the purpose of these different models is. The reader can easily get lost here.

RESPONSE: We appreciate your suggestions and questions to improve the comprehensiveness of the models used and respond to each comment in the following:

(1) Regarding convergence:

When maximum likelihood estimation with robust standard errors (MLR) failed to converge—likely due to model complexity or limited sample size—we used Bayesian estimation as an alternative as this approach has been shown to improve convergence and parameter stability in complex latent variable models, particularly when traditional estimation methods encounter difficulties (B. Muthén & Asparouhov, 2012; Ulitzsch et al., 2023). This approach was also taken in other high quality papers (e.g., Scharbert et al., 2024). Moreover, Bayesian estimation was necessary to meaningfully interpret null results regarding age differences, as Bayesian estimation allows for probabilistic statements about parameter estimates and supports the evaluation of evidence in favor of the absence of an effect. On page 19 we elaborate on these model specifications for convergence issues.

"For Bayes estimation, we employed 10,000 iterations per analysis for explicit self-concepts and 20,000 for implicit self-concepts and models including latent interactions to achieve convergence with values below 1.1 of the Gelman–Rubin diagnostic (Potential Scale Reduction Factor, PSRF (Gelman & Rubin, 1992; B. Muthén & Asparouhov, 2012). To verify estimation accuracy, we used the first half of iterations as a burn-in, to ensure that estimates and PSRF values remained consistent when doubling iterations"

And displayed model fits (now with clear reference to each model) on pp. 12-13 of the Supplementary Material:

Supplementary Table S7

Model Fit Indices of Latent Difference Models (Model A)

Model	χ^2	CFI	TLI	RMSEA	SRMR
A: Latent neighbor change					
ES explicit	43.985	0.992	0.990	0.037	0.052
EX explicit	37.642	0.999	0.998	0.017	0.070
ES implicit	6.666	1.000	1.000	0.000	0.031
EX implicit	5.199	1.000	1.000	0.000	0.018

Note. ES = Emotional stability, EX = Extraversion. CFI = Comparative fit index; TLI = Tucker-Lewis index; RMSEA = Root mean square error of approximation; SRMR = Standardized root mean square residual.

Supplementary Table S8

Model Fit Indices of Latent Growth Models with Bayes Estimator (Model B)

Model	CFI	TLI	RMSEA	BIC
ES explicit	0.971	0.971	0.042	5573.280
EX explicit	1.000	1.000	0.000	4244.147
ES implicit	0.992	0.992	0.025	4966.628
EX implicit	1.000	1.000	0.000	4001.943

Note. ES = Emotional stability, EX = Extraversion. CFI = Comparative fit index; TLI = Tucker-Lewis index; RMSEA = Root mean square error of approximation; BIC = Bayesian information criteria.

Supplementary Table S9

Model Fit Indices of Follow-Up Analyses (Model C)

Model	CFI	TLI	RMSEA	BIC
ES explicit	1.000	1.000	0.000	2772.628
EX explicit	0.940	0.938	0.082	2519.236
ES implicit	1.000	1.000	0.000	739.120
EX implicit	1.000	1.000	0.000	1069.906

Note. ES = Emotional stability, EX = Extraversion. CFI = Comparative fit index; TLI = Tucker-Lewis index; RMSEA = Root mean square error of approximation; BIC = Bayesian information criteria.

(2) Regarding latent modeling and method factors:

Using a latent variable approach with indicator-specific method factors, we followed state-of-the-art recommendations for modeling true change free from measurement error (Geiser & Lockhart, 2012; McArdle, 2009). This approach enables meaningful interpretation of trait change over time while accounting for method effects in a theoretically grounded and psychometrically robust manner. In contrast, manifest approaches risk conflating true trait variance with measurement error and method bias, undermining validity (McArdle, 2009). Specifically, correlated uniqueness models may confound stable trait variance with error and tend to underestimate indicator reliability, making them a less suitable alternative (Geiser & Lockhart, 2012). We therefore refrained from using manifest approaches and models without explicit method factors, as they fall short of the conceptual and empirical standards required for accurate trait modeling in longitudinal research.

(3) Regarding main effects of age:

As shown in Table 1 (see #R2_4), younger and older adults did not differ meaningfully in personality traits at T1, which is plausible given the self-selected nature of the sample. We included this information now (see (4)).

(4) Regarding improving the clarity and understanding of models:

In line with the reviewer's suggestions, we revised the model description for more comprehensiveness and detail (pp. 16- 19):

"To control for item-specific method variance, we included indicator-specific method factors (IS2 and IS3) for the second and third trait parcels, respectively (Figure 2). These factors captured residual covariation among parcels across time points. They were specified with

fixed loadings of 1 (i.e., constrained for identification) and constrained to be uncorrelated with the trait and growth factors, ensuring that only method-specific variance—rather than trait-related variance—was modeled. This modeling strategy is often preferred over specifying correlated residuals because it provides a more parsimonious and psychometrically robust approach to controlling for item-specific method effects (Geiser & Lockhart, 2012).

As no baseline differences in age were observed in personality states and traits, we did not include age group as a predictor of intercept factors. However, age group was included as a predictor of slope factors to test whether the magnitude of change differed across age groups.

Model A was a latent neighbor change model estimating trait change across two neighboring time intervals (T1 to T2 and from T2 to T3). This model tested whether changes differed between the first and second intervention phases, while accounting for their temporal proximity and dependence (Figure 3A). We used this approach to first examine a finer temporal resolution of change, which was especially relevant for extraversion because it was targeted in the second phase of the intervention (i.e., training sessions 5 to 8, corresponding to the time between T2 to T3). Age group was included as a predictor of trait change during each phase. We used the maximum likelihood estimator (MLR) with robust standard errors. Model fit indices of all latent neighbor change models were good and are displayed in Supplementary Table S7.

Model B was a bivariate latent growth model designed to test whether state changes predicted trait changes continuously during the intervention (see Figure 3B; (Muthén & Muthén, 1998-2017)). Each model included an intercept and a growth factor. The intercept factor loadings were set to 1 across time, meaning the intercept represents the baseline level. Latent slope loadings were defined such that each unit increment represented one week. For the trait slope, loadings were set to 0, 4, and 8, corresponding to the approximately 4-week intervals between the three trait assessments (T1, T2, and T3). For the state slope, loadings were set to 0 through 7 across the eight weekly state assessments. The trait slope was predicted by the state slope, age group, and their interaction. Model fit indices of all latent growth models were good and are displayed in Supplementary Table S8.

Model C was a piecewise latent growth model used to examine long-term trajectories of trait change following the intervention, meaning whether changes were maintained, amplified, or diminished. For that purpose, change was modeled separately for the intervention phase (T1 to T3) and the follow-up phase (i.e., T3 to T5). Accordingly, we specified one intercept factor with factor loadings set to 1 across time, and two latent growth factors (see Figure 3C): Slope

1 captured change across the intervention phase and a plateau thereafter, while Slope 2 captured potential change during the follow-up period. Slope loadings were specified as 3 and 12 to represent intervals of 3 and 12 months post-intervention. Both slopes were predicted by age group. Model fit indices of all piecewise latent growth models were good and are displayed in Supplementary Table S9. The model for explicit self-concept of extraversion showed good fit without age but demonstrated poor fit when age was included."

#R2_4: Shouldn't there be a table with the descriptives (means, SDs) for the trait and state variables somewhere? And by age group? Table S15 has correlations, but no means (SDs).

RESPONSE: Table 1 and Table 2 included means, SDs, and N for trait and state measurements, respectively. Following your suggestion, we now present the values separately for each age group on page 20:

Table 1

Descriptive Statistics of Explicit and Implicit Self-Concepts of Emotional Stability and Extraversion by Age Group

Variable M (SD) n	T1	T2	T3	T4	T5
Younger Adults					
Emotional Stability					
Explicit	2.86 (0.56) 80	3.08 (0.62) 74	3.22 (0.60) 68	3.28 (0.60) 62	3.24 (0.59) 48
Implicit	0.31 (0.36) 73	0.32 (0.38) 69	0.26 (0.46) 63		0.25 (0.48) 46
Extraversion					
Explicit	3.21 (0.67) 80	3.36 (0.64) 74	3.44 (0.59) 68	3.41 (0.46) 62	3.29 (0.49) 48
Implicit	-0.22 (0.49) 73	-0.06 (0.51) 69	-0.04 (0.46) 63		-0.02 (0.50) 45
Older Adults					
Emotional Stability					
Explicit	2.85 (0.65) 81	3.06 (0.66) 77	3.28 (0.64) 73	3.36 (0.63) 61	3.35 (0.74) 52
Implicit	0.36 (0.37) 70	0.36 (0.35) 72	0.29 (0.36) 72		0.45 (0.33) 48
Extraversion					
Explicit	3.21 (0.60) 81	3.31 (0.64) 77	3.37 (0.60) 73	3.30 (0.55) 61	3.29 (0.59) 52
Implicit	-0.20 (0.62) 69	-0.13 (0.60) 72	-0.10 (0.63) 72		-0.02 (0.73) 48

Table 2

Descriptive Statistics of State Emotional Stability and Extraversion by Age Group

Variable	Week 1	Week 2	Week 3	Week 4	Week 5	Week 6	Week 7	Week 8
M (SD)								
n								
Younger Adults								
	4.04	4.21	4.08	4.33	4.47	4.59	4.79	4.66
Emotional	(1.20)	(1.00)	(1.43)	(1.70)	(1.63)	(1.56)	(1.71)	(1.77)
Stability	76	73	72	72	70	66	67	63
	4.56	4.70	4.76	4.82	4.95	4.93	5.06	5.04
Extra-	(1.25)	(1.00)	(1.09)	(1.40)	(1.29)	(1.42)	(1.14)	(1.05)
version	76	73	72	72	70	66	67	63
Older Adults								
	4.26	4.27	4.46	4.28	4.49	4.64	4.70	4.93
Emotional	(1.51)	(1.68)	(1.63)	(1.57)	(1.79)	(1.70)	(1.83)	(1.80)
Stability	78	81	79	75	79	78	76	71
	4.82	4.80	4.97	4.81	4.99	5.07	5.08	5.13
Extra-	(1.05)	(1.29)	(1.17)	(1.23)	(1.04)	(1.01)	(1.07)	(1.10)
version	78	81	79	75	79	77	76	72

Minor issues:

#R2_5: Define ES and EX and include dfs if chi-square values are reported (Table S5, S8, S10, S16).

RESPONSE: We followed the reviewer's suggestion to define ES and EX. Dfs are not reported in Bayesian estimation. Bayesian analysis avoids traditional frequentist test statistics like the t-test, which rely on sampling distributions and their degrees of freedom (see Kruschke, 2013). Instead, Bayesian inference directly estimates posterior distributions for parameters. Accordingly, we reported the posterior SDs instead (see Supplementary Tables S12, S13, S15).

#R2_6: Figure 4 looks as if it was resized unevenly; the axes font looks strange. Redo the font and maybe make the legend slightly larger.

#R2_7: Include different markers/lines (e.g., dashed) for the four groups in Figure 4 to improve accessibility for people with vision problems.

RESPONSE: We appreciate the helpful suggestions regarding Figure 4 and have modified it accordingly. Specifically, we adjusted the font size and proportions to improve readability and used different line styles and markers to enhance accessibility for individuals with visual impairments (p.24):

Figure 4 *Change in Explicit and Implicit Self-Concepts of Emotional Stability and Extraversion During the Personality Intervention and at 3- and 12-months Follow-Up*

Note. The figure displays the changes in standard deviation units for the outcome variables between the baseline (T1) and further time points. Implicit measures were not assessed at T4.

#R2_8: In Table S11, isn't it obvious that there is a difference? Does it really need subscripts?

RESPONSE: We agree and deleted the subscripts.

Respectfully yours,

References

- Allan, J., Leeson, P., Fruyt, F. D., & Martin, S. (2018). *Application of a 10 week coaching program designed to facilitate volitional personality change: Overall effects on personality and the impact of targeting*. <https://doi.org/10.24384/000470>
- Allredge, C. T., Burlingame, G. M., Yang, C., & Rosendahl, J. (2021). Alliance in group therapy: A meta-analysis. *Group Dynamics: Theory, Research, and Practice*, 25(1), 13–28. <https://doi.org/10.1037/gdn0000135>
- Allemand, M., Olaru, G., Stieger, M., & Flückiger, C. (2024). Does realizing strengths, insight, and behavioral practice through a psychological intervention promote personality change? An intensive longitudinal study. *European Journal of Personality*, 08902070231225803. <https://doi.org/10.1177/08902070231225803>
- Back, M. D., Schmukle, S. C., & Egloff, B. (2009). Predicting actual behavior from the explicit and implicit self-concept of personality. *Journal of Personality and Social Psychology*, 97(3), 533–548. <https://doi.org/10.1037/a0016229>
- Baumert, A., Schmitt, M., Perugini, M., Johnson, W., Blum, G., Borkenau, P., Costantini, G., Denissen, J. J. A., Fleeson, W., Grafton, B., Jayawickreme, E., Kurzius, E., MacLeod, C., Miller, L. C., Read, S. J., Roberts, B., Robinson, M. D., Wood, D., & Wrzus, C. (2017). Integrating Personality Structure, Personality Process, and Personality Development. *European Journal of Personality*, 31(5), 503–528. <https://doi.org/10.1002/per.2115>
- Behn, A., Davanzo, A., & Errázuriz, P. (2018). Client and therapist match on gender, age, and income: Does match within the therapeutic dyad predict early growth in the therapeutic alliance? *Journal of Clinical Psychology*, 74(9), 1403–1421. <https://doi.org/10.1002/jclp.22616>
- Bleidorn, W., Schwaba, T., Zheng, A., Hopwood, C. J., Sosa, S., Roberts, B., & Briley, D. A. (2022). *Personality Stability and Change: A Meta-Analysis of Longitudinal Studies* [Preprint]. PsyArXiv. <https://doi.org/10.31234/osf.io/eq5d6>
- Borghuis, J., Bleidorn, W., Sijtsma, K., Branje, S., Meeus, W. H. J., & Denissen, J. J. A. (2019). *Longitudinal associations between trait neuroticism and negative daily experiences in adolescence*. <https://doi.org/10.31234/osf.io/d3bym>

- Borghuis, J., Bleidorn, W., Sijtsma, K., Branje, S., Meeus, W. H. J., & Denissen, J. J. A. (2020). Longitudinal associations between trait neuroticism and negative daily experiences in adolescence. *Journal of Personality and Social Psychology, 118*(2), 348–363.
<https://doi.org/10.1037/pspp0000233>
- Carstensen, L. L. (1995). Evidence for a Life-Span Theory of Socioemotional Selectivity. *Current Directions in Psychological Science, 4*(5), 151–156. <https://doi.org/10.1111/1467-8721.ep11512261>
- Chaney, K. E., & Chasteen, A. L. (2024). Do Beliefs That Older Adults Are Inflexible Serve as a Barrier to Racial Equality? *Personality and Social Psychology Bulletin, 50*(8), 1151–1166.
<https://doi.org/10.1177/01461672231159767>
- Danner, D., Rammstedt, B., Bluemke, M., Treiber, L., Berres, S., Soto, C., & John, O. (o. J.). *Die deutsche Version des Big Five Inventory 2 (BFI-2)*. 21.
- Finkelstein, L. M., Burke, M. J., & Raju, M. S. (1995). Age discrimination in simulated employment contexts: An integrative analysis. *Journal of Applied Psychology, 80*(6), 652–663.
<https://doi.org/10.1037/0021-9010.80.6.652>
- Fleeson, W., & Gallagher, P. (2009). The implications of Big Five standing for the distribution of trait manifestation in behavior: Fifteen experience-sampling studies and a meta-analysis. *Journal of Personality and Social Psychology, 97*(6), 1097–1114. <https://doi.org/10.1037/a0016786>
- Gawronski, B., & Bodenhausen, G. V. (2006). Associative and propositional processes in evaluation: An integrative review of implicit and explicit attitude change. *Psychological Bulletin, 132*(5), 692–731. <https://doi.org/10.1037/0033-2909.132.5.692>
- Gawronski, B., & LeBel, E. P. (2008). Understanding patterns of attitude change: When implicit measures show change, but explicit measures do not. *Journal of Experimental Social Psychology, 44*(5), 1355–1361. <https://doi.org/10.1016/j.jesp.2008.04.005>
- Geiser, C., & Lockhart, G. (2012). A comparison of four approaches to account for method effects in latent state–trait analyses. *Psychological Methods, 17*(2), 255–283.
<https://doi.org/10.1037/a0026977>

- Gelman, A., & Rubin, D. B. (1992). Inference from Iterative Simulation Using Multiple Sequences. *Statistical Science*, 7(4). <https://doi.org/10.1214/ss/1177011136>
- Greenwald, A. G., Nosek, B. A., & Banaji, M. R. (2003). Understanding and using the Implicit Association Test: I. An improved scoring algorithm. *Journal of Personality and Social Psychology*, 85(2), 197–216. <https://doi.org/10.1037/0022-3514.85.2.197>
- Hampson, S. E. (2012). Personality Processes: Mechanisms by Which Personality Traits “Get Outside the Skin”. *Annual Review of Psychology*, 63(1), 315–339. <https://doi.org/10.1146/annurev-psych-120710-100419>
- Hofmann, W., Gawronski, B., Gschwendner, T., Le, H., & Schmitt, M. (2005). A Meta-Analysis on the Correlation Between the Implicit Association Test and Explicit Self-Report Measures. *Personality and Social Psychology Bulletin*, 31(10), 1369–1385. <https://doi.org/10.1177/0146167205275613>
- Kruschke, J. K. (2013). Bayesian estimation supersedes the t test. *Journal of Experimental Psychology: General*, 142(2), 573–603. <https://doi.org/10.1037/a0029146>
- Lang, F. R., & Carstensen, L. L. (1994). Close emotional relationships in late life: Further support for proactive aging in the social domain. *Psychology and Aging*, 9(2), 315–324. <https://doi.org/10.1037/0882-7974.9.2.315>
- Lücke, A. J., Quintus, M., Egloff, B., & Wrzus, C. (2020). You can't always get what you want: The role of change goal importance, goal feasibility and momentary experiences for volitional personality development. *European Journal of Personality*, 089020702096233. <https://doi.org/10.1177/0890207020962332>
- Mader, N., Arslan, R. C., Schmukle, S. C., & Rohrer, J. M. (2023). Emotional (in)stability: Neuroticism is associated with increased variability in negative emotion after all. *Proceedings of the National Academy of Sciences*, 120(23). <https://doi.org/10.1073/pnas.2212154120>
- Maxwell, S. E., Kelley, K., & Rausch, J. R. (2008). Sample Size Planning for Statistical Power and Accuracy in Parameter Estimation. *Annual Review of Psychology*, 59(1), 537–563. <https://doi.org/10.1146/annurev.psych.59.103006.093735>

- McArdle, J. J. (2009). Latent Variable Modeling of Differences and Changes with Longitudinal Data. *Annual Review of Psychology, 60*(1), 577–605.
<https://doi.org/10.1146/annurev.psych.60.110707.163612>
- Muthén, B., & Asparouhov, T. (2012). Bayesian structural equation modeling: A more flexible representation of substantive theory. *Psychological Methods, 17*(3), 313–335.
<https://doi.org/10.1037/a0026802>
- Muthén, L. K., & Muthén, B. (1998, 2017). *Mplus user's guide (8th ed.)*.
- Quintus, M., Egloff, B., & Wrzus, C. (2021). Daily life processes predict long-term development in explicit and implicit representations of Big Five traits: Testing predictions from the TESSERA (Triggering situations, Expectancies, States and State Expressions, and ReActions) framework. *Journal of Personality and Social Psychology, 120*(4), 1049–1073.
<https://doi.org/10.1037/pspp0000361>
- Rauthmann, J. F. (2024). Personality is (so much) more than just self-reported Big Five traits. *European Journal of Personality, 38*(6), 863–866.
<https://doi.org/10.1177/08902070231221853>
- Roberts, B. W., Luo, J., Briley, D. A., Chow, P. I., Su, R., & Hill, P. L. (2017). A systematic review of personality trait change through intervention. *Psychological Bulletin, 143*(2), 117–141.
<https://doi.org/10.1037/bul0000088>
- Scharbert, J., Humberg, S., Kroencke, L., Reiter, T., Sakel, S., Ter Horst, J., Utesch, K., Gosling, S. D., Harari, G., Matz, S. C., Schoedel, R., Stachl, C., Aguilar, N. M. A., Amante, D., Aquino, S. D., Bastias, F., Bornamanesh, A., Bracegirdle, C., Campos, L. A. M., ... Back, M. D. (2024). Psychological well-being in Europe after the outbreak of war in Ukraine. *Nature Communications, 15*(1). <https://doi.org/10.1038/s41467-024-44693-6>
- Smillie, L. D., Wilt, J., Kabbani, R., Garratt, C., & Revelle, W. (2015). Quality of social experience explains the relation between extraversion and positive affect. *Emotion, 15*(3), 339–349.
<https://doi.org/10.1037/emo0000047>

- Soto, C. J., & John, O. P. (2017). The next Big Five Inventory (BFI-2): Developing and assessing a hierarchical model with 15 facets to enhance bandwidth, fidelity, and predictive power. *Journal of Personality and Social Psychology, 113*(1), 117–143.
<https://doi.org/10.1037/pspp0000096>
- Stieger, M., Flückiger, C., Rügger, D., Kowatsch, T., Roberts, B. W., & Allemand, M. (2021). Changing personality traits with the help of a digital personality change intervention. *Proceedings of the National Academy of Sciences, 118*(8), e2017548118.
<https://doi.org/10.1073/pnas.2017548118>
- Suls, J., & Martin, R. (2005). The Daily Life of the Garden-Variety Neurotic: Reactivity, Stressor Exposure, Mood Spillover, and Maladaptive Coping. *Journal of Personality, 73*(6), 1485–1510.
<https://doi.org/10.1111/j.1467-6494.2005.00356.x>
- Tamm, J., Takano, K., Just, L., Ehring, T., Rosenkranz, T., & Kopf-Beck, J. (2024). Ecological Momentary Assessment versus Weekly Questionnaire Assessment of Change in Depression. *Depression and Anxiety, 2024*(1). <https://doi.org/10.1155/2024/9191823>
- Ulitzsch, E., Lüdtke, O., & Robitzsch, A. (2023). Alleviating estimation problems in small sample structural equation modeling—A comparison of constrained maximum likelihood, Bayesian estimation, and fixed reliability approaches. *Psychological Methods, 28*(3), 527–557.
<https://doi.org/10.1037/met0000435>
- Van Dieren, Q., Rijckmans, M. J. N., Mathijssen, J. J. P., Lobbestael, J., & Arntz, A. R. (2013). Reducing no-show behavior at a community mental health center. *Journal of Community Psychology, 41*(7), 844–850. <https://doi.org/10.1002/jcop.21577>
- Van Zalk, M. H. W., Nestler, S., Geukes, K., Hutteman, R., & Back, M. D. (2020). The codevelopment of extraversion and friendships: Bonding and behavioral interaction mechanisms in friendship networks. *Journal of Personality and Social Psychology, 118*(6), 1269–1290.
<https://doi.org/10.1037/pspp0000253>

- Wesch, D., Lutzker, J. R., Frisch, L., & Dillon, M. M. (1987). Evaluating the impact of a service fee on patient compliance. *Journal of Behavioral Medicine, 10*(1), 91–101.
<https://doi.org/10.1007/bf00845130>
- Wrenn, K. A., & Maurer, T. J. (2004). Beliefs About Older Workers' Learning and Development Behavior in Relation to Beliefs About Malleability of Skills, Age-Related Decline, and Control¹. *Journal of Applied Social Psychology, 34*(2), 223–242. <https://doi.org/10.1111/j.1559-1816.2004.tb02546.x>
- Wrzus, C., Luong, G., Wagner, G. G., & Riediger, M. (2021). Longitudinal coupling of momentary stress reactivity and trait neuroticism: Specificity of states, traits, and age period. *Journal of Personality and Social Psychology, 121*(3), 691–706. <https://doi.org/10.1037/pspp0000308>
- Wrzus, C., & Roberts, B. W. (2017). Processes of Personality Development in Adulthood: The TESSERA Framework. *Personality and Social Psychology Review, 21*(3), 253–277.
<https://doi.org/10.1177/1088868316652279>